# Recent Progress of Basic Studies of Natural Products and Their Dental Application

**DOI:** 10.3390/medicines6010004

**Published:** 2018-12-25

**Authors:** Hiroshi Sakagami, Taihei Watanabe, Tomonori Hoshino, Naoto Suda, Kazumasa Mori, Toshikazu Yasui, Naoki Yamauchi, Harutsugu Kashiwagi, Tsuneaki Gomi, Takaaki Oizumi, Junko Nagai, Yoshihiro Uesawa, Koichi Takao, Yoshiaki Sugita

**Affiliations:** 1Meikai University Research Institute of Odontology (M-RIO), 1-1 Keyakidai, Sakado, Saitama 350-0283, Japan; 2Division of Pediatric Dentistry, Meikai University School of Dentistry, 1-1 Keyakidai, Sakado, Saitama 350-0283, Japan; taiheidental10627@gmail.com (T.W.); thoshino1@dent.meikai.ac.jp (T.H.); 3Division of Orthodontics, Meikai University School of Dentistry, 1-1 Keyakidai, Sakado, Saitama 350-0283, Japan; n-suda@dent.meikai.ac.jp; 4Division of First Oral and Maxillofacial Surgery, Meikai University School of Dentistry, 1-1 Keyakidai, Sakado, Saitama 350-0283, Japan; kazu-mori@dent.meikai.ac.jp; 5Division of Oral Health, Meikai University School of Dentistry, 1-1 Keyakidai, Sakado, Saitama 350-0283, Japan; yasui@dent.meikai.ac.jp; 6Masuko Memorial Hospital, 35-28 Takehashi-cho, Nakamura-ku, Nagoya 453-8566, Japan; yamauchi@masuko.or.jp; 7Ecopale Co., Ltd., 885 Minamiisshiki, Nagaizumi-cho, Suntou-gun, Shizuoka 411-0932, Japan; ecopale@fujibamboogarden.com; 8Gomi clinic, 1-10-12 Hyakunin-cho, Shinjuku-ku, Tokyo 169-0073, Japan; fwkz9633@mb.infoweb.ne.jp; 9Daiwa Biological Research Institute Co., Ltd., 3-2-1 Sakado, Takatsu-ku, Kawasaki, Kanagawa 213-0012, Japan; takaakio@daiwaseibutsu.co.jp; 10Department of Medical Molecular Informatics, Meiji Pharmaceutical University, 2-522-1 Noshio, Kiyose, Tokyo 204-8588, Japan, nagai-j@my-pharm.ac.jp (J.N.); uesawa@my-pharm.ac.jp (Y.U.); 11Department of Pharmaceutical Sciences, Faculty of Pharmacy and Pharmaceutical Sciences, Josai University, Sakado, Saitama 350-0295, Japan; ktakao@josai.ac.jp (K.T.); sugita@josai.ac.jp (Y.S.)

**Keywords:** polyphenol, chromone, lignin-carbohydrate complex, alkaline extract, Kampo medicine, glucosyltransferase, angiotensin II blocker, QSAR analysis, oral diseases, dental application

## Abstract

The present article reviews the research progress of three major polyphenols (tannins, flavonoids and lignin carbohydrate complexes), chromone (backbone structure of flavonoids) and herbal extracts. Chemical modified chromone derivatives showed highly specific toxicity against human oral squamous cell carcinoma cell lines, with much lower toxicity against human oral keratinocytes, as compared with various anticancer drugs. QSAR analysis suggests the possible correlation between their tumor-specificity and three-dimensional molecular shape. Condensed tannins in the tea extracts inactivated the glucosyltransferase enzymes, involved in the biofilm formation. Lignin-carbohydrate complexes (prepared by alkaline extraction and acid-precipitation) and crude alkaline extract of the leaves of *Sasa* species (SE, available as an over-the-counter drug) showed much higher anti-HIV activity, than tannins, flavonoids and Japanese traditional medicine (Kampo). Long-term treatment with SE and several Kampo medicines showed an anti-inflammatory and anti-oxidant effects in small size of clinical trials. Although the anti-periodontitis activity of synthetic angiotensin II blockers has been suggested in many papers, natural angiotensin II blockers has not yet been tested for their possible anti-periodontitis activity. There should be still many unknown substances that are useful for treating the oral diseases in the natural kingdom.

## 1. Introduction

The etiology of stomatitis is largely unclear [1]. However, oral inflammation such as stomatitis are considered to be triggered or aggravated by various factors including bacterial and viral infections, nutritional deficiencies, declined immune functions, allergic reactions, radiotherapy, stress, cigarettes, diseases and genetic backgrounds [1,2]. Applications of topical steroids, transdermal patches, vitamins, throat lozenges, mouth washes and cryotherapy are sometimes not effective for the treatment of stomatitis and therefore exploration of new-type of treatment are necessary [3]. In this sense, natural products having broader spectrum of biological activities are potential candidates as alternative medicine for oral diseases.

Polyphenols in the natural kingdom are defined as substances that possess an aromatic ring bearing one or more hydroxyl substituents and roughly classified into tannins, flavonoids and lignin-carbohydrate complexes (LCC) (Figure 1) [4]. 

Tannins are further classified into hydrolysable tannins (in which a polyalcohol is esterified with a galloyl, hexahydroxydiphenoyl, valoneoyl or dehydrohexahydroxydiphenoyl group) and condensed tannins (composed of catechin, epicatechin or their analogs) (Figure 1A) [5]. 

Flavonoids, synthesized from chalcones [6], are classified into flavonols, flavones, flavanones and isoflavones (that contain the chromone structure in the molecule), pterocarpan and coumestan (Figure 1B). Due to the recent development of separation technology [7,8], chemical structures and biological functions of thousands of tannins and flavonoids have been elucidated. 

Lignin is formed by dehydrogenative polymerization of *p*-coumaryl, *p*-conifery and sinapyl alcohols and forms a complex with some polysaccharides (Figure 1C). Lignin-carbohydrate complex (LCC) has amorphous structure with very high molecular weight, thus making it difficult to determine the complete chemical structure, although it shows prominent anti-HIV activity [9]. Since LCC can be prepared by alkaline solution and acid-precipitation, it was not surprising that alkaline extract of the leaves of *Sasa* species (*Sasa* sp.) (SE) described later contains significant amount of LCC and shows several over-lapped biological activities with LCC.

It is generally accepted that improvement of oral functions by periodontal treatment [10], insertion of dentures and implants [11], oral hygiene [12], nutrition [13] and fluoride treatment [14] elevates the general health and quality of life [10,11]. Orally administered products directly contact the oral tissues or cells where they may exert their effects very fast, without being metabolizing and excretion [15], if they have a chance to bind to the target molecules or pattern-recognition receptors such as TLR2 (Toll-like receptor 2), TLR4, Dectin-1 (receptor for glucan) and Dectin-2 (receptor for LCC or mannan) in keratinocytes, macrophages, monocytes and dendritic cells [16]. This article reviews the recent progress of three major polyphenols (tannins, flavonoids and LCCs), chromone (backbone structure of flavonoids) and herbal extracts, glucosyltransferase inhibitor and angiotensin II blocker on dental diseases. 

## 2. Chromone Derivatives as New Type of Anticancer Candidate 

### 2.1. Most of Anticancer Drugs Show Severe Keratinocyte Toxicity 

Development of anticancer drugs is shifting from classical anti-cancer drugs to molecular targeted therapeutic agents. However, the incidence of complete response in gastroesophageal cancer patients treated with targeted agents has been reported to be 2.0%, only 0.3 increase from the control arms [17]. ErbB receptor-targeting inhibitors failed to show any significant differences on overall response rate, clinical benefit rate and overall survival, with the increased risk of serious adverse events [18]. Likewise, cyclin-dependent kinase inhibitor combined with chemotherapy slightly increased the mean progression-free survival but also stimulated the senescence-associated (SA) marker expression (assessed by the accumulation of by SA β-galactosidase in the lysosome) by yet unknown mechanism [19]. This points out another unfavorable effect of targeted therapy, the resolution of which we have to find urgently. 

Administration of anticancer agents has been reported to induce skin toxicity [20,21,22,23,24,25,26]. This prompted us to re-evaluate the cytotoxicity and tumor-specificity of anticancer drugs. We demonstrated for the first time that classical anticancer drugs (doxorubicin, daunorubicin, etoposide, mitomycin C, methotrexate, 5-fluorouracil, melphalan) and molecular targeted therapeutic drug (gefinitib) are highly toxic to epithelial normal cells (keratinocytes) as well as human oral squamous cell carcinoma (OSCC) cell lines. Tumor specificity (TS), determined with human normal oral epithelial cells *vs* OSCC cells (TS_E_ = 0.1 to 1.5) was usually one to two-orders lower than TS, determined with mesenchymal normal cells *vs* OSCC cells (TS_M_ = 3.8 to 92.9) [27] (Exp. 1, Table 1).

Also, doxorubicin induced apoptosis characterized by chromatin condensation, nuclear fragmentation and loss of cell surface microvilli) (A) and caspase-3 activation (cleavage of PARP and pro-caspase-3) (B) in human oral keratinocytes [27] (Figure 2). This urged us to survey many natural products which show lower keratinocyte toxicity.

### 2.2. Limitations of Apoptosis-Oriented Research

Many studies have reported the apoptosis-inducing activity of tannins and flavonoids but have not tested for their toxicity to normal cells or tumor selectivity. We reevaluated the antitumor effect of various groups of natural products, based on the TS values determined as shown in the insert of Figure 3. As expected, anticancer drugs showed excellent tumor-specificity (TS_M_, determined by the ratio of mean CC_50_ for human normal oral mesenchymal cells to that for human OSCC cell lines, indicated by red color). We found that one among14 poly-herbal formula extracts (supplied by Himalaya drug company) showed excellent tumor-specificity [28]. The active principle (s) are yet to be determined. It was surprising that the tumor selectivity of flavonoids, procyanidins, macrocyclic ellagitannins, hydrolysable tannins, catechins and gallic acid, which has been reported to induce apoptosis, was surprisingly low (TS_M_ = 1 to 5) (green color), as compared with anticancer agents. Similarly, antioxidants (vitamin C, chlorogenic acid, curcumin), ketones (α,β-unsaturated ketones, α-hydroxyketones, β-diketones, trifluoromethylketones, zulenequinones) and amides (pheylpropanoid amides, piperic acid amides, oleoylamides) showed lower TS_M_ values. On the other hand, the tumor selectivity of eight chromone derivatives (A–H) described later was relatively high (yellow color) (Figure 3).

### 2.3. Synthesis of Chromone Derivatives Having High Tumor-Specificity and Low Keratinocyte Toxicity

Chromone (4*H*-1-benzopyran-4-one) is a backbone structure of flavonols, flavones, flavanones and isoflavones [29] (Figure 1B). We synthesized eight classes of chromones derivatives (total 134 compounds): 3-styrylchromones (15 compounds) [30,31] (containing compound A), 3-benzylidenechromanones (17 compounds) [32] (containing compound B), 3-styryl-2*H*-chromenes (16 compounds) [33] (containing compound C), 2-azolylchromones (24 compounds) [34] (containing compound D), 3-(*N*-cyclicamino)chromones (15 compounds) [35] (containing compound E), 2-(*N*-cyclicamino)chromones (15 compounds) [36] (containing compound F), furo[2,3-*b*]chromones (12 compounds) [37] (containing compound G) and pyrano[4,3-*b*]chromones (20 compounds) [38] (containing compound H). The eight compounds that produced the highest TS value in each group are listed in Figure 4.

All compounds showed much higher cytotoxicity against human oral squamous cell carcinoma (OSCC) cell lines (Ca9-22, HSC-2, HSC-3, HSC-4) than against human normal oral mesenchymal cells (gingival fibroblast HGF, periodontal ligament fibroblast HPLF, pulp cell HPC). These compounds except for 3-benzylidenechromanones were 2.6~2000-fold less cytotoxic to human oral keratinocytes as compared with doxorubicin (Exp. 2, Table 1). We reported that 3-styrylchromones [30] and azolylchromones [34] induced apoptosis (caspase-3 activation) in human OSCC cell line. On the contrary, 7-methoxy-2-(4-morpholinyl)-4*H*-1-benzopyran-4-one, the most active compound among fifteen 2-(*N*-cyclicamino)chromone derivatives (structure depicted in Figure 4F) showed an excellent tumor-specificity (TS = 63.4) (Figure 5A), low keratinocyte toxicity (Table 1, Exp. 2), without induction of apoptosis in human OSCC cell line (HSC-2), as evidenced by the lack of caspase-3 activation (cleavage of PARP and procaspase-3) (Figure 5B) nor of the accumulation of subG_1_ population (Figure 5C).

In order to perform the QSAR analysis with each group of compounds, the 3D structure of each chemical structure was optimized by CORINA Classic (Molecular Networks GmbH, Nürnberg, Germany) with forcefield calculations (amber-10: EHT) in Molecular Operating Environment (MOE) version 2018.0101 (Chemical Computing Group Inc., Quebec, Canada). Approximately 3000 chemical descriptors were analyzed for their correlation with cytotoxicity against tumor cells (**T**) and normal cells (**N**) and tumor-specificity (**T–N**), suggesting that molecular shape is the most important determinant for tumor-specificity (Table 2). For example, we have reported previously that **T–N** of 3-styrylchromones can be estimated by diameter (largest value in the distance matrix defined by the elements Dij), vsurf_DD23 and R3 OH (n = 15, R^2^ = 0.764, Q^2^ = 0.570, s = 0.308) (right), according to the following equation: **T–N** = 0.607(± 0.169)diameter – 0.121 (± 0.035)vsurf_DD23 + 1.11 (± 0.235)R3OH – 7.17 (± 2.26) [30]. QSAR analysis can be applied to estimate the most potent chemical structures. By repeating the process of synthesis of the estimated structure and reconfirmation of its activity, more active compounds with defied structure will be manufactured.

Metabolomic analysis is powerful to determine the early event of cell death induction process. We have reported that compound A (which induced apoptosis) increased the intracellular levels of diethanolamine and CDP-choline and reduced that of choline, suggesting the down-regulation of the glycerophospholipid pathway [31]. It remains to be determined which metabolic pathway is first affected at early stages after treatment with compound F (which did not induce apoptosis). 

## 3. Catechins as Inhibitors of Glucosyltransferase

### 3.1. Classification of Oral Streptococcal GTF Enzymes

Dental plaque is the oral biofilm that consists of bacteria themselves and bacterial metabolites. Glucan, polymer of glucose, is one of the metabolically-produced polysaccharides as the basic structures of the dental plaque and is produced from sucrose by glucosyltransferase enzymes (GTFs). Since dental plaque is a fertile ground of the pathogenic bacteria and virus that cause oral disease such as stomatitis, dental caries, gingivitis and periodontitis, glucan and/or GTFs are the pathogen of those diseases. These GTFs are produced mainly by streptococci in oral cavity [39,40]. 

Oral streptococcal GTFs (EC: 2.4.1.5) [41] are encoded by *gtf* genes, belong to the glycosyl hydrolase family 70 and basically catalyze the transfer of D-glucopyranosyl units from sucrose to acceptor molecules [42]. Biochemically, GTFs are classified into mainly 2 types according to their products, water-soluble glucan (WSG) and water-insoluble glucan (WIG), main components of oral biofilm. Especially in *Streptococcus mutans (S. mutans)*, GTFB, water-insoluble glucan synthesizing glucosyltransferase enzyme is one of virulence factors for dental caries, because water-insoluble glucan plays an important role in adhesion and establishment of *S. mutans* on tooth surface [43,44,45]. 

To clarify the ancestry of streptococcal GTFs, we investigated the distribution of GTFs among bacteria, such as *Lactobacillus*, *Leuconostoc* and *Lactococcus* and phylogenetically analyzed glycosyl hydrolase family 70 enzymes [46]. The sequences of glycosyl hydrolase family 70 proteins used in this study were obtained from GenBank at NCBI (http://www.ncbi.nlm.nih.gov/) with reference to Pfam (http://pfam.sanger.ac.uk/). Sequence alignment was performed using ClustalW software version 1.83 [47] (http://clustalw.ddbj.nig.ac.jp/index.php?, DNA Data Bank of Japan, Mishima, Japan). Multiple alignment files saved by ClustalW in Clustal format were converted to MEGA format with the MEGA version 5 software [48] (http://www.megasoftware.net/). Phylogenetic analysis was performed by the maximum parsimony methods using MEGA version 4.0 software. We analyzed 20 glucosyltransferases from *Streptococcus*; 2 glucosyltransferases, 9 dextran sucrases and 1 alternan sucrase from *Leuconostoc*; 10 glucan sucrases from *Lactobacillus*; and 1 glucosyltransferases from *Lactococcus*. PspA, glucosyltransferases from *Lactococcus* was defined as the convenient ancestor in this analysis (Figure 6) [46].

The accession numbers in NCBI of these glycosyl hydrolases are provided after species and enzyme name in Figure 6. Here, we have shown that enzymes in the lower part of the phylogenetic tree synthesize glucans with various linkage types such as α-1,3; α-1,6; α-1,2; and α-1,4, while those in the upper part of the tree synthesize only water-soluble α-1,6-linked glucans. The phylogenetic tree would suggest that the streptococcal GTFs were derived from other lactic acid bacteria following their spread through the genera in the order of *Lactococcus*, *Lactobacillus*, *Leuconostoc* and *Streptococcus* and that the streptococcal GTF family can be phylogenetically classified into 3 clusters: the water-soluble glucan-synthesizing group (WSG), the water-insoluble glucan synthesizing group (WIG) and the intermediate group (INT).

A phylogenetic tree was constructed using the maximum parsimony (MP) method. The value on each branch is the estimated confidence limit (expressed as a percentage) for the position of the branches, as determined by bootstrap analysis. Only values exceeding 50% are shown. [46]

### 3.2. Purification of GTF Enzymes

Streptococcal GTFs can be divided into 2 types, WIG- and WIG- synthesizing GTFs by glucan product. Native WIG-synthesizing GTF, for example, GTFB and GTFC was purified from *S. mutans* MT8148, cultured in TTY medium [49,50]. The bacteria were collected by centrifugation and the cell-associated GTF (CA-GTF) was extracted by 8 M urea. The extract was precipitated by 60% saturated ammonium sulfate, applied to a DEAE Sepharose FF and eluted with a linear gradient of 0 to 1.0 M NaCl in the same buffer. Active fractions measured were concentrated by ammonium sulfate precipitation and then applied to a Bio-Scale CHT10-I column and then eluted with a 10 to 500 mM potassium phosphate buffer (KPB) linear gradient [51]. To select the GTFB and GTFC fractions, we carried out glucan synthesis assay [52], ELISA using anti-CA-GTF antibody, SDS-PAGE and Western blot using anti-GTF-I (GTFB) monoclonal antibody and anti-GTF-SI (GTFC) monoclonal antibody (Figure 7) [49]. 

Native WSG-synthesizing GTF, for example, GTFR was purified from *Streptococcus oralis* (*S. oralis*) ATCC 10557 [53], cultured in dialyzed TTY medium [50]. The culture supernatant was precipitated by 60% saturated ammonium sulfate and then applied to a Q Sepharose FF column and eluted with a linear gradient of 0 to 1.0 M NaCl [51]. Active fractions (indicated by bars) were pooled, applied to a Bio-Scale CHT10-I column and then eluted with a 10 to 500 mM KPB linear gradient [51]. To select the GTFR, the glucan synthesis assay and SDS-PAGE were carried out (Figure 8) [53].

In summary, purification of oral streptococcal GTFs would be commonly carried out by three steps method, ammonium sulfate precipitation, ion-exchange chromatography and hydroxyapatite chromatography [54,55,56,57,58]. Recombinant GTF (rGTF) would be produced as the other approach of GTF preparation using expression vector [59,60,61,62,63]. rGTF would be purified by ammonium sulfate precipitation, chromatography using His-tag and so on. These native and/or recombinant GTFs were used for the suppression analysis of glucan production by inhibitor and the recovery analysis of biofilm formation by addition of GTF. For these reasons, preparation of oral streptococcal GTFs would be important in the study of prevention of oral infectious diseases such as dental caries, periodontitis and so on.

### 3.3. Inhibitors of Oral Streptococcal GTFs

In GTFR from *S. oralis*, some inorganic salts suppressed the synthesis of water-soluble glucan. The synthesis of water-soluble glucan was reduced especially by divalent cation. Therefore, divalent cation could be inhibitors. However, in water-insoluble GTF from *Streptococcus sobrinus* 6715, high concentrations of monovalent (above 100 mM) and divalent (above 20 mM) cations stimulated the formation of insoluble glucan, whereas lower concentrations of monovalent (below 10 mM) and divalent (below 1 mM) cations reduced the formation of insoluble glucan to a negligible amount [64]. Thus, it would be difficult to adopt inorganic salts an inhibitor of oral streptococcal GTFs, considering their opposing actions between high and low concentrations.

It has been well known that natural products, for example, hydrolysable tannins (gallotannin, ellagitannin), condensed tannins (proanthocyanin, catechins), complex tannins, as from plant origin such as green tea, Oolong tea, cocoa, coffee and traditional Chinese medicine, inhibit glucan synthesis of oral streptococcal GTFs [65,66,67,68,69,70,71]. Especially, polyphenol mixtures from Oolong tea or cacao beans among them inhibited glucan-synthesis activity of GTFs from *S. mutans*. For example, OTF6, one of polyphenol fraction extracted from Oolong tea inhibited glucan-synthesis activity of rGTFB, rGTFC and rGTFD (Figure 9) [72]. With the increase of substrate of GTF, the production of glucan reached the plateau (near saturation) level. Even if the substrate concentration is enough, OTF6 effectively inhibited the production of glucan, suggesting its application to the dental plaque and caries. Since they can be inhibitors against other diseases, for example, stomatitis, periodontitis and aspiration pneumonia, they are expected to inhibit the formation of various glucan-biofilm, which contains some pathogenic organisms [73].

Nearly half of the commensal bacterial population of the human body is present in the oral cavity. An increase in the number of oral microorganisms may produce infective endocarditis, aspiration pneumonia and oral infections. When hydroxypropylcellulose strips containing green tea catechin were applied once a week for 8 weeks in pockets as a slow release local delivery system, the patient’s periodontal status was significantly improved [74]. Gel-entrapped catechin (GEC) was manufactured by mixing catechins (epigallocatechin, epigallocatechin gallate, epicatechin, epicatechin gallate, gallocatechin, catechin and gallocatechin gallate) with polysaccharide, dextrin, citric acid, potassium chloride and stevia, to maintain the moistness in the oral cavity of elderly patients. GEC inhibited the growth of the Actinomyces, periodontopathic bacteria and Candida strains, possibly due to the produced hydrogen peroxide [75]. Local treatment of GEC seems to be important, since orally-administered catechin have been reported to increase the blood mitochondrial heme amounts and catalase activity, that may neutralize the antimicrobial activity of GEC [76].

## 4. Lignin-Carbohydrate Complex (LCC) as Anti-HIV Resources of the Natural Kingdom

We have previously reported anti-HV activity of three major polyphenols, tannins, flavonoids and lignin-carbohydrate complex (LCC), that were purified by our group. The potency of anti-HIV activity (SI) was calculated from the following equation: SI = CC_50_/EC_50_, where the CC_50_ is the concentration that reduced the viable cell number of the uninfected cells by 50% and the EC_50_ is the concentration that increased the viable cell number of the HIV-infected cells up to 50% that of the control (mock-infected, untreated) cells. Among them, LCC from pine cones of *Pinus parviflora* Sieb. et Zucc, pine cone of *Pinus elliottii var*. Elliottii, pine seed shell of *Pinus parviflora* Sieb. et Zucc, bark of *Erythroxylum catuaba* Arr. Cam, husk and mass of cacao beans of Theobroma, *Lentinus edodes* mycelia extract (L·E·M) and from precipitating fiber fraction of mulberry juice [77,78,79,80,81,82,83,84] showed the highest value (SI = 14, 28, 12, 43, 311, 46, 94 and 7), although much lower than that of popular anti-IIV agents (dextran sulfate, curdlan sulfate, azidothymidine, 2′,3′-dideoxycytidine) (SI = 2956 to 23261) (Table 3). Lignin but not carbohydrate moiety, seems to be essential to exert the anti-HIV activity, since synthetic lignin, manufactured by dehydrogenation polymerization of phenylpropenoids showed the comparable anti-HIV activity [85], whereas neutral and uronic acid-containing polysaccharides were inactive (SI = 1) [86]. We also found that monomer of phenylpropanoid monomers (*p*-coumaric acid, ferulic acid, caffeic acid) were inactive (SI < 1) [85], suggesting the importance of higher-ordered complicated structures for anti-HIV activity induction.

On the other hand, both hydrolysable and condensed tannins (see Figure 1A for classification) [87] (SI = 1.8 to 7.3 and 1.1) and flavonoids (Figure 1B) [88] (SI = 1.5) showed much lower anti-HIV activity. It is noted that anti-HIV activity of hydrolysable tannins increased with degree of polymerization: monomer (SI = 1.8) < dimer (SI = 2.3) < timer (SI = 3.4) < tetramer (SI = 7.3) [87]. 

Alkaline extraction of green tea leaves, oolong tea leaves, orange flower, licorice root was more efficient than hot water extraction to recover the anti-HIV substances: SI = 3 *vs* < 0.022; 13 *vs* < 0.033; > 15 *vs* < 0.5; 42 *vs* 4, respectively [89,90]. Likewise, alkaline extract of leaves of *Sasa* sp. showed much higher anti-HIV activity (SI = 86) than Japanese traditional medicines, Kampo (SI = 1.0) and constituent plant extracts (SI = 1.3) [91]. Chromone, such (*E*)-3-(4-hydroxystyryl)-6-methoxy-4*H*-chromen-4-one and (*E)*-3-(4-Chlorostyryl)-7-methoxy-2*H*-chromene were inactive [30] (Table 3).

## 5. Alkaline Extract of the Leaves of *Sasa* sp. (SE)

### 5.1. Prominent Anti-HIV, Anti-UV, Anti-Inflammation and Neuroprotective Activities (in vitro)

Although alkaline extracts of plants showed much higher anti-HIV activity than corresponding hot water extracts [86,89,90,91], only three papers from other groups have reported the anti-angiogenic and neuroprotective activity of alkaline extracts [92,93,94]. Also, only two papers have been published on the isolation and fractionation of lignin from bamboo, however, they reported no data of biological activity [95,96]. Based on these backgrounds, we reviewed mostly our research topics of SE.

Alkaline extract of the leaves of *Sasa* sp. (SE) is an over-the counter (OTC) drug in Japan, which is available in the drug store without the prescription of doctors. SE (dry weight: 58.8 mg/mL) contains Fe (II)-chlorophyllin, LCC and its degradation products and so forth. SE showed higher anti-HIV [86], anti-UV [97,98], anti-inflammatory [99] and neuroprotection activities [100], as compared with other lower molecular polyphenols (Table 4). SE has many good partners for exerting synergistic actions: anti-HIV activity with azidothymidine, 2′,3′-dideoxycytidine, dextran sulfate or curdlan sulfate [101]; anti-HSV activity with acyclovir [101], anti-bacterial activity with isopropyl methylphenol [102] and anti-UV activity [103] and radical scavenging activity with vitamin C [104]. SE also showed osteogenic activity [105]. 

Among three SE products, product A (100% pure SE that contains Fe(II)-chlorophyllin) showed 1~5-fold higher anti-HIV, anti-UV and hydroxyl radical scavenging activity and 3~7-fold lower CYP3A4 inhibitory activity than product B (contain Cu(II)-chlorophyllin and less LCC) and product C (product B further supplemented with ginseng and *Pinus densiflora* leaf extracts) [106]. Based on this finding, we used product A for the following studies and manufacturing the toothpaste.

### 5.2. Improvement of Lichenoid Dysplasia by SE

Oral *lichen planus* is a chronic mucocutaneous disease that affects tongue and oral mucosa, characterized by white lacy streaks on the mucosa or as smaller papules. The cause of lichen planus is not known. A biopsy was taken from a 43-year-old male patient and diagnosed as lichenoid dysplasia in 7 July 7 2003 (physician in charge: Dr. K. Mori). Treatment with vitamin B_1_ improved the patient’s symptoms but discontinuation of the treatment resulted in the disease recurrence. The patient was subjected to the SE treatment for 11 months (12 April 2011 until 12 March 2012), according to the guideline of Intramural Ethic Committee (no. A0901). The patient was orally administered 13.3 ml SE (diluted two-fold with water, thus containing 33 mg dried materials/mL) three times-a-day, 30 min before each meal. At each administration, the patient swallowed and retained SE in the oral cavity for 1 min before washing it down. The patient did not take any other medications during the treatment period. The patient’s oral cavity was photographed with a digital camera and the total saliva was collected just before lunch and then every two weeks, after the start of SE administration.

When a patient had been treated for 12 months with SE, white areas of lacy streaks in the several areas of buccal mucosa progressively reduced (Figure 10A). Oral intake of SE also improved the patient’s symptoms of pollen allergy and loose teeth, giving an impression that the oral mucosa became much tighter. Three weeks after treatment, uneven, rough and cut mucosa became much smoother. At four weeks, the rough mucosa was narrowed into a smaller area and the patient could eat without any pungent feeling on the oral mucosa. SE treatment reduced the salivary concentration of IL-6 from 0.052 ± 0.030 ng/mL (n = 5) to 0.01 ng/mL and that of IL-8 from 5.25 ± 1.06 ng/mL (n = 5) to 1.11 ng/mL [107]. 

### 5.3. Anti-Oxidative Stress Effect of SE in Chronic Dialysis Patients

With the cooperation of 10 maintenance dialysis patients in Masuko Memorial Hospital for 2 years from 2000, clinical data of SE were accumulated (Figure 11A–E). 

By treatment with SE, superoxide (O^2-^) was gradually declined. Although the drift was seen in the middle of 12 months, possibly due to the measuring problem with instruments and after 24 months, cases exceeding the initial level were not observed (A). Likewise, LPO showed a decreasing tendency, possibly due to the elimination of O^2-^, although no significant difference was observed (B). In support of this finding, the SOD activity gradually increased (C), suggesting the enhancement of the antioxidant action. Concentration of markers for the impairment of vascular endothelium (thrombomodulin and von Willebrand factor) did not show any significant fluctuation (measured by SRL Inc.), with large variation of the data due to small numbers of patients [108].

A shunt, which is a bypass connecting the artery and vein of the brachium to perform blood maintenance dialysis, is indispensable. However, the pressure due to the dialysis causes expansion, the bending and extension of blood vessel, the thickening of vein wall and the ectopic calcification due to the repeated puncture, leading to the blockage due to the decreased blood pressure, stagnation of blood flow and enhancement of coagulability. All patients frequently repeat the clotting in the part of shunt during dialysis. However, the incidences of such shunt troubles were apparently reduced by SE treatment (D) and questionnaire on the clinical effects of SE showed the good outcome (E) (Figure 11). Although this clinical trial has ended once in 2 years, one patient (84 years old) still receives the clinical trial, with little or no trouble of the shunt, possibly due to the preservation of blood fluidity and maintenance of blood vessel possibly by iron chlorophyllin and antioxidants in SE [108].

### 5.4. Anti-Halitosis Effect of Toothpaste Supplemented with SE

Considering the potent anti-HIV, anti-inflammatory effects of SE, we have manufactured SE containing toothpaste (SETP) for the first time. The SETP is composed of 26.2% SE, 0.1% isopropylmethylphenol, base materials, cleaning agents, humectants, flavoring substances, sweetening agent, stabilizers, binding agent and washing soap). We have selected the 26.2% SE, since treatment of periodontal ligament fibroblasts for 1 min with 50% of SE did not affect the cell viability and approximately 8 ml of saliva were produced and accumulated in the oral cavity by 5 min of tooth brushing. SETP can be obtained at the drug store. We investigated its anti-halitosis effect with the collaboration of a total of 12 volunteers, according to the guideline of Intramural Ethic Committee (no. A1219). They brushed their teeth immediately after meals three times each day with STEP or placebo toothpaste (omitting only SE). Halitosis in the breath and bacterial number on the tongue were measured by portable apparatuses at 11:00 AM in the morning. We found that SETP significantly reduced halitosis (*p* = 0.046) but not the number of bacterial on the tongue (*p* = 0.60) [109].

### 5.5. Other Unpublished Case Reports 

The female subject (28 years old) was bothered by atopy since her infancy. When her living environment changed by getting a job three years ago, the symptoms worsened further especially in early spring, rainy season and dry winter. She unexpectedly found that pasting the *Sasa* sp. extract-immersed cotton on her skin improved the symptoms. In another when red eczema and skin roughness became apparent (A), she applied just a pearl-size amount of “Moisture Creamy Gel” (containing to 1.8% *Sasa* sp. extract), her skin of cheek and face line three times a day. The creamy gel produced no bleeding, in contrast to other commercially available lotions. After 1 week, area of red eczema began to diminish and rough skin became smooth, thus reducing the application time to twice a day. After 2 weeks, eczema has completely healed (B) (Figure 12).

Recently, diacetyl (2,3-butanedione), mostly produced by *Staphylococcus aureus* and *Staphylococcus epidermidis*, has been reported to be a key contributor to unpleasant odors emanating from the axillae, feet and head regions [110]. There was one case report that when male healthy volunteer (69 years old) take daily drink of SE (40 mL, 2.4 g dry weight) mixed with lemon and orange juice after lunch except for Sunday, he experienced the significant reduction in the fecal smell (after 1 week), tongue cloth (after 1 month) and body odor (after three months) and no stress-induced stomatitis for 7 months. 

## 6. Kampo Medicines

Xerostomia is a disease in which a reduction in salivary secretion causes oral dryness and it may also be further complicated with odontonecrosis, periodontal disease, candidiasis, and taste disorders. The exacerbation of these diseases has a substantial effect on the QOL, so it is necessary in such cases to clarify the causes and select the most appropriate treatment. Herbal treatment alleviated thirst and oral dryness in most cases but many cases showed a slower increase in salivary production than the cases administered cevimeline hydrochloride [111] (Figure 13).

When the patient with xerostomia-induced glossitis was treated with Byakkokaninjinto (3 g three times a day) for two months, glossitis largely disappeared, and the subjective symptoms decreased (Figure 14).

## 7. Dental Application of Angiotensin II Receptor Blocker for Severe Periodontitis

### 7.1. Angiotensin II Receptor Blocker (ARB) in Marfan Syndrome

Marfan syndrome is an autosomal dominant connective tissue disease that affects about one in 5000 individuals [113]. The responsible gene of this syndrome is FBN1 which encodes the extracellular matrix protein fibrillin-1 [113]. FBN1 mutations lead to defects in multiple organs including skeletal, cardiovascular and ocular systems [114]. Among them, the most serious problems are seen in the cardiovascular system, such as, aortic regurgitation, aneurysm and dissection of the aortic root, mitral valve prolapse and mitral regurgitation, causing a short life expectancy in patients [115]. 

Fibrillin-1 regulates the function of endogenous transforming growth factor (TGF)-β by targeting the respective complexes to the extracellular cell matrix [116]. Studies of animal [117] and human [118] reported that TGF-β signaling drives aneurysm progression in the aorta [119]. Since Marfan syndrome patients have cardiovascular problems, the surgical replacement of aortic and mitral valve and aortic roots is often required [115]. It is known that effects of angiotensin II are mediated by two receptors, type 1 (AT1) and type 2 (AT2) receptor [120]. AT1-receptor signaling can increase the production of TGF-β ligands and receptors [121]. Angiotensin II-receptor blockers (ARBs) selectively block the binding of angiotensin II to its receptor within the renin–angiotensin system [122]. AT1-receptor blockade decreases TGF-β signaling and thus inhibit the phosphorylation of Smad2. Recently, losartan, one of ARBs, have been reported to suppress the progression of aortic root dilation by inhibiting TGF-β signaling [117,123]. Application of ARBs are now providing great benefit to Marfan syndrome patients by improving cardiovascular conditions.

### 7.2. Periodontal Disease Frequently Seen in Marfan Syndrome

Oral manifestations are not included as diagnostic criteria of Marfan syndrome but this disease is frequently affected with severe periodontitis [124,125,126]. Periodontitis affects periodontal tissues, including gingiva, periodontal ligament (PDL) and alveolar bone [127]. Approximately 15% of the adult population has an advanced form of periodontitis, making multiple negative impacts on quality of life [128,129]. Consequences of periodontitis include negative esthetics and functional problems in occlusion, chewing and speaking and finally result in tooth loss [130,131]. Periodontitis is initiated by chronic inflammation and immune reactions to bacterial pathogens [132]. Several bacteria play important roles in the pathogenesis of periodontitis but *Porphyromonas gingivalis* plays a central role in pathogenesis of periodontitis [133]. It is reported that 87.5% of Marfan syndrome patients had periodontitis with more than 4 mm of periodontal pocket depth, while only 35.7% of healthy volunteers showed such manifestation [134]. Interestingly, higher percentage of periodontitis with more than 4 mm of periodontal pocket depth was seen in patients with cardiovascular disease than those without cardiovascular disease [135]. Many Marfan syndrome patients have these cardiovascular problems, often necessitating the surgical replacement of aortic and mitral valve and aortic roots [115]. Because of this surgical replacement, it is essential to prevent dental infection, such as infectious endocarditis caused by the periodontitis. 

The reason of higher incidence of severe periodontitis in Marfan in not known. However, the lower number of caries has been reported in adult Marfan syndrome patients than in healthy volunteers [124]. This implies that periodontal tissues but not teeth have structural problems making susceptible to severe periodontitis. The abnormal alignment of collagen fibers was observed in one of the model mice of Marfan syndrome (MgR mice). Homozygous MgR mice show the 72% of reduction in Fbn1 (encoding mouse fibrillin-1) expression because of transcriptional interference by insertion of the PGKneo-cassette [136] and resemble the phenotype of Marfan syndrome by showing 10% longer long bones than wild-type (WT) littermates. A comparable level of type I collagen, which is the most major collagen in periodontal ligaments, was expressed in PDL-cells of homozygous MgR mice as in WT mice [137]. However, multi-oriented collagen fiber bundles with a thinner appearance were noted in homozygous mice. These observations were never seen in WT mice showing well-organized definite collagen fiber bundles. This suggests that normal level of fibrillin-1 is essential for the normal architecture of periodontal ligament. 

### 7.3. Progression of Periodontal Disease and Application of ARB

Telmisartan (4′-[[4-methyl-6-(1-methyl-1*H*-benzimidazol-2-yl)-2-propyl-1*H*-benzimidazol-1-yl]methyl]biphenyl-2-carboxylic acid) is a non-peptide ARBs used in the management of hypertension [138,139] and expected as an effective drug for the management of vascular condition in Marfan syndrome [140]. This drug has a binding affinity 3,000 times higher for AT1 than AT2 [141]. Heterozygous MgΔ mice, another mice model of Marfan syndrome, show half level of Fbn1 as WT mice [136]. Six-week-old male heterozygous Mg∆ and WT mice were challenged with *P. gingivalis* with and without telmisartan application [142]. Infection of *P. gingivalis* induced alveolar bone resorption in both heterozygous MgΔ and wild-type mice. The amount of alveolar bone resorption was significantly larger in the former than the latter. Interleukin (IL)-17 and tumor necrosis factor (TNF)-α levels were significantly higher in infected Mg∆ mice than infected WT mice. Telmisartan treatment significantly suppressed the alveolar bone resorption of infected Mg mice. Telmisartan also significantly reduced the levels of TGF-β, IL-17 and TNF-α in infected Mg∆ mice to those seen in infected WT mice. These suggest that ARB can prevent the severe periodontitis frequently seen in Marfan syndrome. Combination with Chinese medicine and angiotensin-converting enzyme inhibitors (ACEI) or ARB showed kidney protection effect [143,144] and tannic acid inhibited AT1 gene expression and cellular response [145]. However, previous studies have not yet investigated whether traditional medicines and dietary polyphenols inhibit the periodontitis through blocking the AT1. 

## 8. Future Direction

The present article demonstrated that chromone derivatives show high tumor-specificity, low keratinocyte toxicity, without or with induction of apoptosis, suggesting that apoptosis induction is not the absolute necessity for exploration of anticancer drugs. Chromone ring is a natural material, distributing into many flavonoids. By introducing an appropriate substituent thereto with the guidance of QSAR analysis, more active derivatives can be manufactured. Synthesis of ^13^C-labled chromone derivatives is underway to investigate the cellular uptake and binding to the specific acceptor molecules in the cells. Since all data of chromone derivatives are produced in vitro, in vivo study with implanted tumors are necessary to confirm the selective action against tumor cells. Also, possibility of synergistic action with anticancer drugs and effects on CYP-3 enzymes that affect the stability of accompanying drugs should be monitored before the clinical application.

LCC and SE, both are extracted from plants by alkaline extraction showed extremely higher anti-HIV, anti-inflammatory and neuroprotective activity. Our recent study demonstrated that SE stimulated the growth of differentiated neuronal cells and human gingival epithelial progenitor (HGEP) at lower concentration. This hormetic action of SE may explain its ability to protect the cells from amyloid peptides. The pathogenesis of dementia is thought to be due to the collapse of cerebral nerve cells and the reduction of neurotransmitters by the senile plaques produced by the accumulation of amyloid beta (Aβ) and tau protein (Tau) in the brain. It remains to be investigated whether SE prevent the dementia, if so by what mechanism. 

We have previously reported that LCC of SE, prepared by repeated acid-precipitation and alkaline solubilization, has greenish color (absorption peak = 655 nm), characteristic to chlorophyllin (absorption peak = 629 nm) and that 68.5% of SE eluted as a single peak at the retention time of 22.175 min in HPLC [146]. This suggests that LCC in SE may easily bind to or entangled with chlorophyllin and other components to make large molecule under physiological condition. Such large molecule may non-specifically bind to many cell surface receptors including dectin-2, causing its unique biological activity. 

This review suggests the efficacy of GTF inhibitors and ARBs to prevent the biofilm formation and periodontitis, respectively. It is crucial to search for these inhibitors and blockers from the natural kingdom and elucidate their action mechanism. 

## Figures and Tables

**Figure 1 medicines-06-00004-f001:**
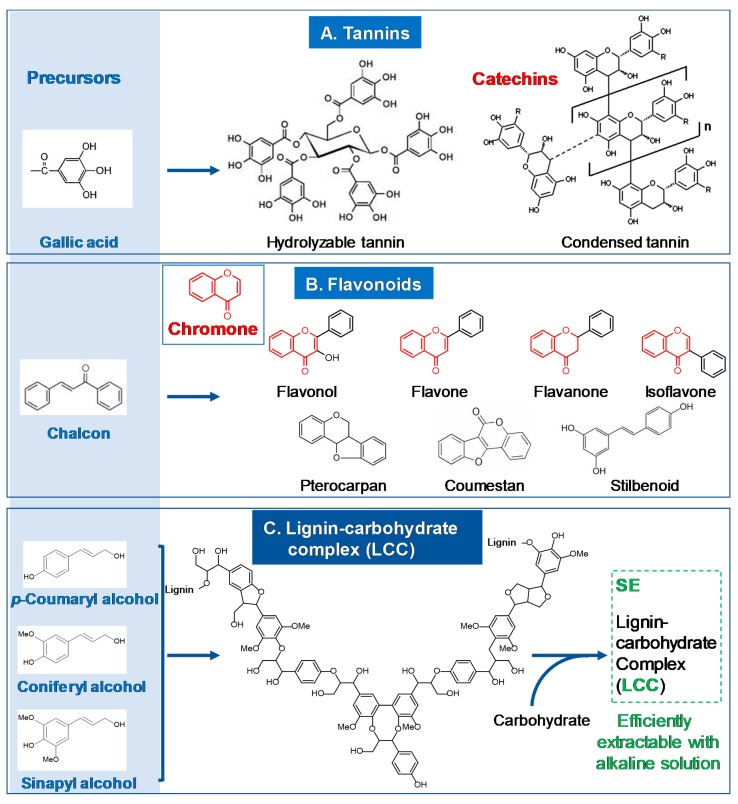
Three major polyphenols, that is tannins (**A**), flavonoids with or with backbone structure of chromone (**B**) and lignin-carbohydrate complex (LCC) (**C**), in the natural kingdom. Cited and modified from Reference [4] with permission.

**Figure 2 medicines-06-00004-f002:**
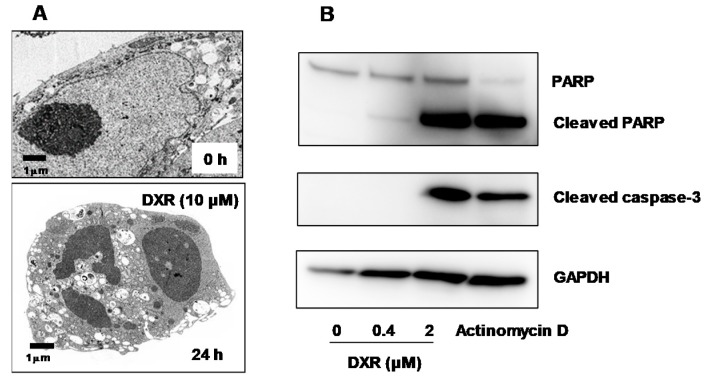
Doxorubicin induced apoptosis in human oral keratinocyte, demonstrated by transmission electron (**A**) and western blot analysis (**B**). Cited from Reference [27] with permission.

**Figure 3 medicines-06-00004-f003:**
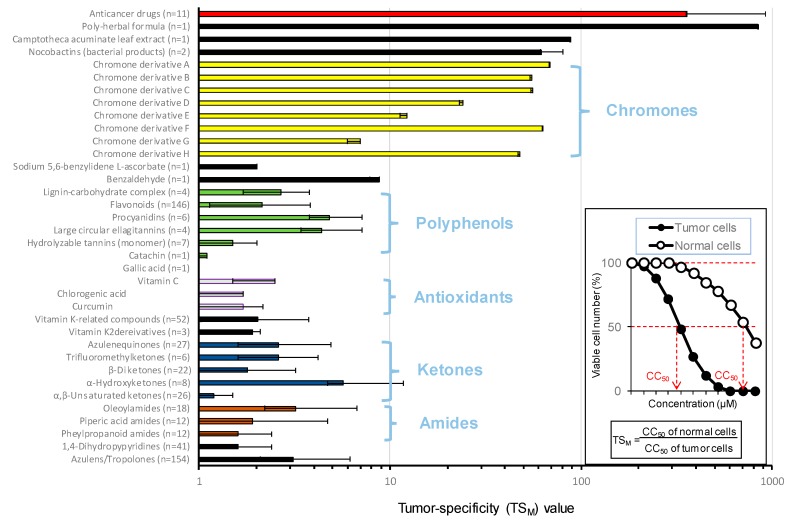
Chromone derivatives showed higher tumor-specificity (TS_M_) value than most of the polyphenols. TS_M_ was determined by the following equation: TS_M_ = (mean CC_50_ for human OSCC cell lines/mean CC_50_ for human normal oral mesenchymal cells. Tumor and normal cells in the insert represent human OSCC cell lines and human normal oral mesenchymal cells. Data of chromones [30,32,33,34,35,36,37,38] and other compounds [4] were cited with permission. n, number of compounds tested.

**Figure 4 medicines-06-00004-f004:**
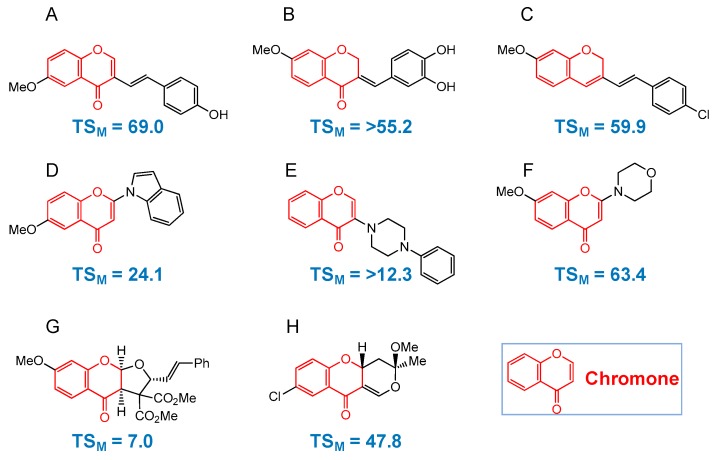
Compounds that showed the highest tumor-specificity (TS_M_) values (determined with human OSCC and human oral mesenchymal cell) in eight groups of chromone derivatives. Structure and TS_M_ values of (**A**) that belongs to 3-styrylchromones [30], (**B**) that belongs to 3-benzylidenechromanones [32], (**C**) that belongs to 3-styryl-2*H*-chromenes [33], (**D**) that belongs to 2-azolylchromones [34], (**E**) that belongs to 3-(*N*-cyclicamino)chromones [35], (**F**) that belongs to 2-(*N*-cyclicamino)chromones [36], (**G**) that belongs to furo[2,3-*b*]chromones [37] and (**H**) that belongs to pyrano[4,3-*b*]chromones [38] were cited with permission.

**Figure 5 medicines-06-00004-f005:**
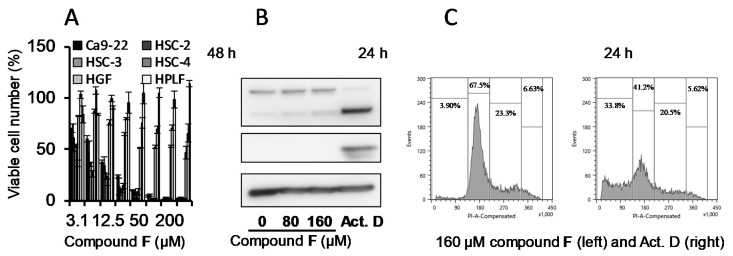
7-Methoxy-2-(4-morpholinyl)-4*H*-1-benzopyran-4-one (Compound F, Figure 4F) showed higher cytotoxicity against human OSCC cell lines as compared with normal oral mesenchymal cells (**A**), without induction of caspase-3 activation (**B**) nor producing subG_1_ cell population (**C**). Actinomycin (Act. D) (1 μM) was used as positive control. Cited from Reference [35] with permission.

**Figure 6 medicines-06-00004-f006:**
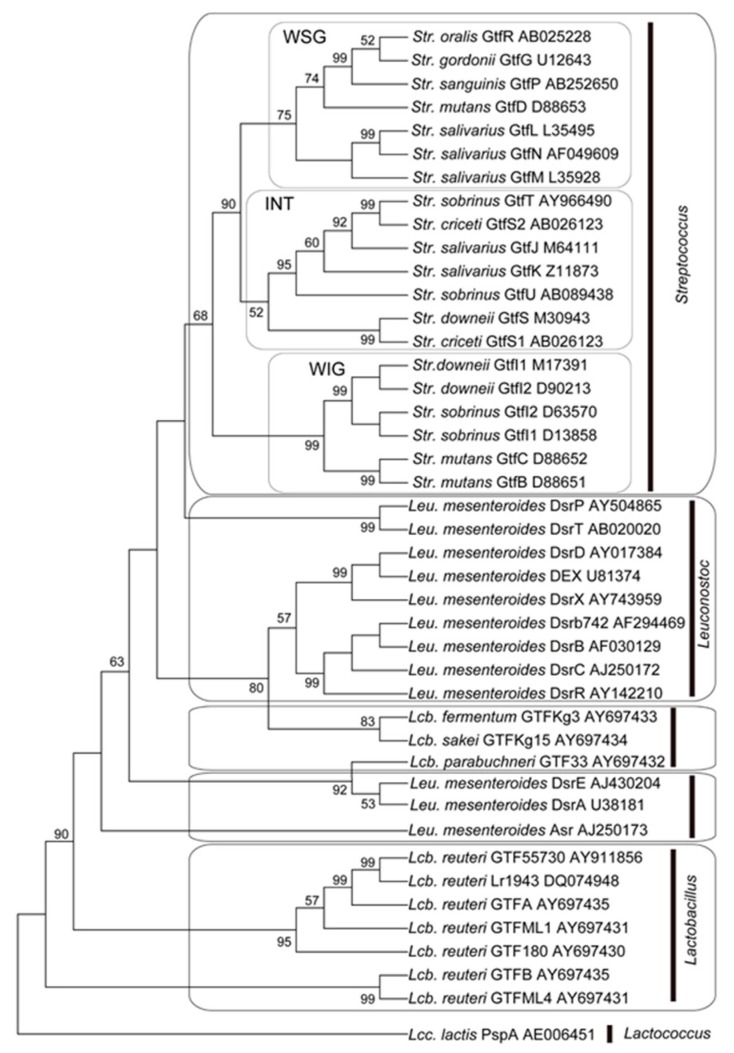
Phylogenetic analysis of glycosyl hydrolase family 70 enzymes by amino acid sequence. Cited from Reference [46] with permission.

**Figure 7 medicines-06-00004-f007:**
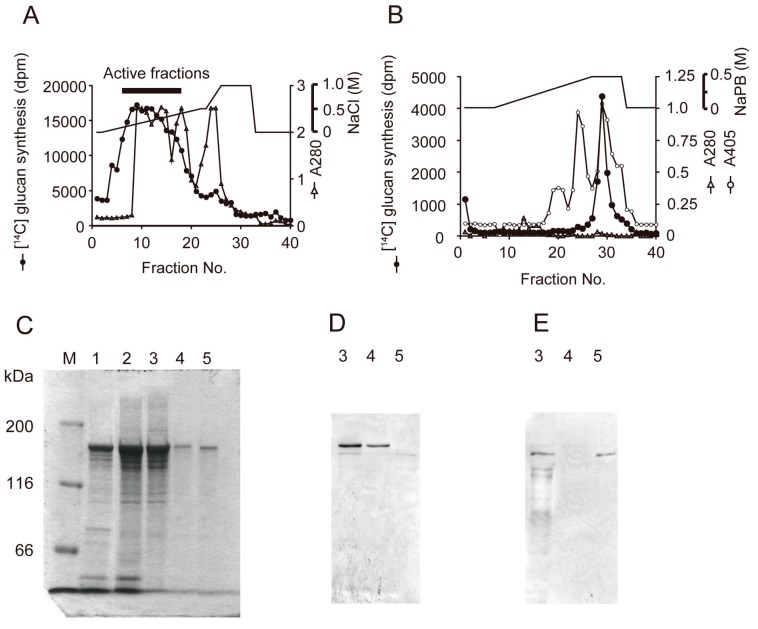
Purification of GTFB and GTFC from *S. mutans* MT8148, by sequential chromatography on. DEAE Sepharose FF column (collecting No. 6 to No. 17 fractions) (**A**) and Bio-Scale CHT10-I column (collecting No.29 and No.24 fraction as purified GTFB and GTFC, respectively) (**B**). The sample in each purification step was separated by SDS-PAGE (**C**) and assessed with Western blot analyses using anti-GTF-I (**D**), anti-GTF-SI (**E**) antiserum. Lane M, molecular weight marker; 1, 8 M-urea extraction; 2, precipitant of 1 by ammonium sulfate; 3, GTF-active fraction eluted with DEAE Sepharose column; 4, No. 29 fraction eluted with CHT10-I column; No. 24 fraction eluted with CHT10-I column. Cited from Reference [49] with permission.

**Figure 8 medicines-06-00004-f008:**
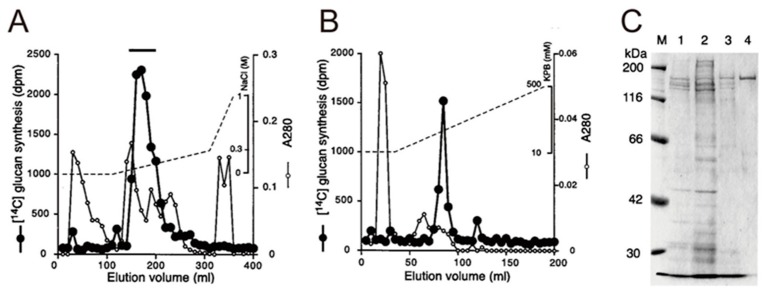
Purification of GTFR from *Streptococcus oralis* (*S. oralis*) ATCC 10557 by sequential chromatography on a Q Sepharose FF column (eluted with a linear gradient of 0 to 0.3 M NaCl) (**A**) and Bio-Scale CHT10-I column (**B**). Elution was done with a 10 to 500 mM KPB linear gradient. (**C**) SDS-PAGE of GTase preparations at different stages of purification. Lanes: 1, culture supernatant; 2, ammonium sulfate precipitate; 3, pooled active fractions from Q-ion-exchange chromatography; 4, pooled active fraction from CHT-10 hydroxylapatite chromatography; M, molecular mass markers. Cited from Reference [53] with permission.

**Figure 9 medicines-06-00004-f009:**
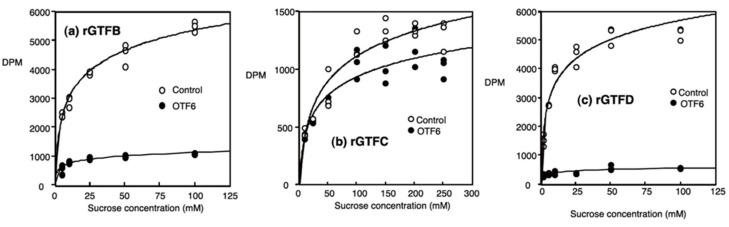
Changes in the quantity of glucan produced by recombinant GTFs. GTF activity was measured with [14C-glucose] sucrose. The OTF6 concentration in all displayed data was 1.0 mg/mL. Data are given in counts per minute. GTFs and sucrose were reacted without (○) and with (●) OTF6. Cited from Reference [72] with permission.

**Figure 10 medicines-06-00004-f010:**
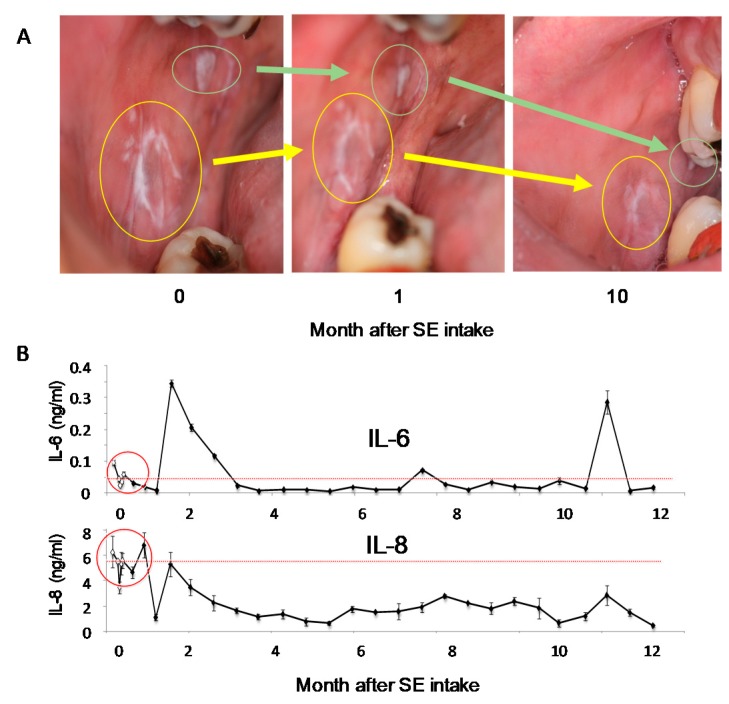
Effect of SE treatment on the oral lichenoid dysplasia. (**A**) Oral inspection with a digital camera, (**B**) salivary IL-6 and IL-8 concentrations. Cited from Reference [107] with permission.

**Figure 11 medicines-06-00004-f011:**
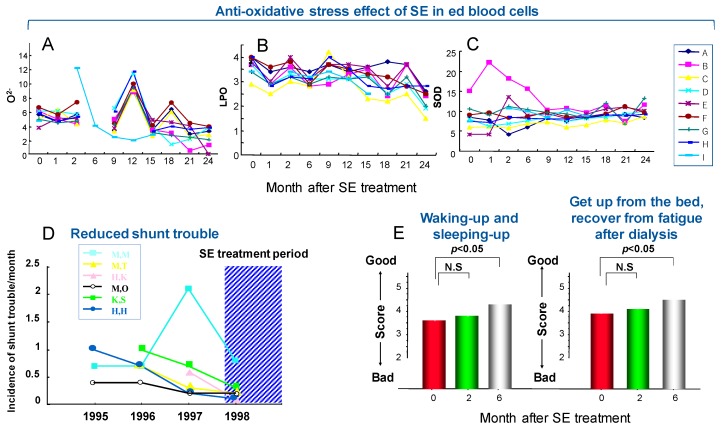
Anti-oxidative stress effects on chronic dialysis patients. (**A**) Superoxide (O^2-^) (measured by luciferin chemiluminescence method), (**B**) LPO (measured by HPLC), (**C**) SOD activity (xanthine ·xanthine oxidase reaction) of red blood cells. (**D**) Incidence of shunt problem from 1996–1998. Patients were treated with SE after 1 November 1997. (**E**) Questionnaire about the clinical effect of SE. Cited from Reference [108] with permission.

**Figure 12 medicines-06-00004-f012:**
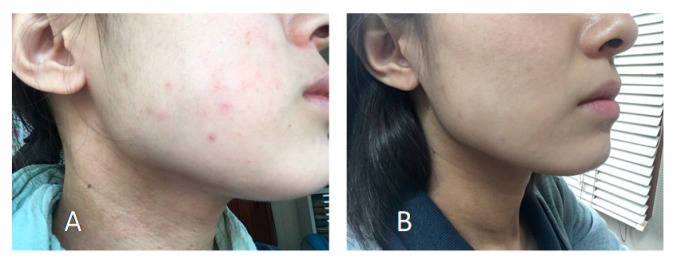
Facial photo before (**A**) and two weeks after application (**B**) of SE Moisture Creamy Gel (taken on 17 April 2018) (unpublished data).

**Figure 13 medicines-06-00004-f013:**
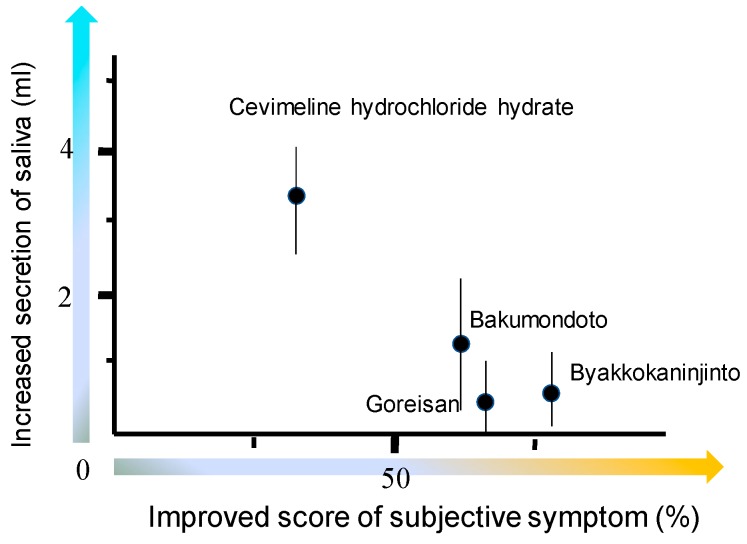
Regarding subjective symptoms, a questionnaire is issued about (i) mouth dryness, (ii) swallowing difficulty and (iii) oral pain, using the VAS method. When the average of these three items is 0, 1~20, 21~40, 41~60, 61~80, 81~100, the score is counted as 1, 2, 3, 4 and 5 points, respectively. The extent of subjective symptom improvement was calculated by dividing the subjective symptom score before the administration by that before starting administration and the multiplied by 100. Cited from Reference [111] with permission.

**Figure 14 medicines-06-00004-f014:**
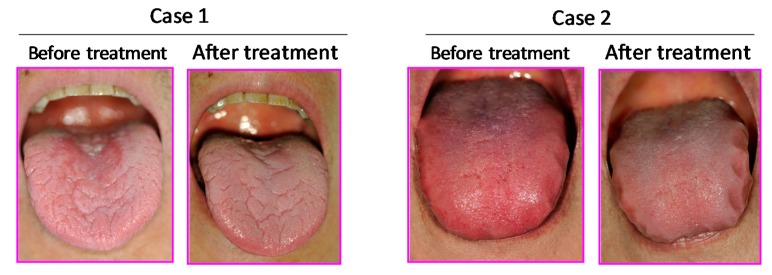
Effect of *Byakkokaninjinto* on glossitis. Cited from Reference [112] with permission.

**Table 1 medicines-06-00004-t001:** Comparison of keratinocyte toxicity between popular anticancer drugs (Exp. 1) and chromone derivatives (Exp. 2).

Compounds	CC_50_ (μM)												TS_M_	TS_E_	
	Human Oral Squamous Cell Carcinoma					Human Oral Normal Cells							Mes	Epi	
						Mesenchymal Cells				Epithelial Cells			vs	vs	
	Ca9-22	HSC-2	HSC-3	HSC-4	mean	HGF	HPLF	HPC	mean	HOK	HGEP	mean	OSCC	OSCC	Ref.
					(A)				(B)			(C)	(B/A)	(C/A)	
**Exp. 1 Anticancer drugs:**															
CPT	<0.06	<0.06	<0.06	<0.06	<0.06	200	10	146	119	0.3	3.9	2.1	**>1853**	**>33**	[27]
SN-38	<0.06	<0.06	<0.06	<0.06	<0.06	143	29	16	63	<0.075	1.5	<0.77	**>979**	**<12**	[27]
DXR	0.13	0.06	0.09	0.06	0.09	7.3	1.3	9.3	6.0	0.1	0.2	0.1	**69.9**	**1.7**	[27]
DNR	0.27	0.07	0.13	0.09	0.14	4.9	10.0	8.2	7.7	<0.004	0.4	<0.21	**54.6**	**<1.5**	[27]
ETP	11.3	3.0	2.7	2.5	4.9	351	500	500	450	1.8	3.2	2.5	**92.9**	**0.5**	[27]
MMC	3.97	0.36	0.14	0.78	1.31	22	65	34	40	0.10	0.28	0.19	**30.8**	**0.1**	[27]
MTX	9.0	0.2	<0.13	<0.13	<2.35	>400	>400	>400	>400	1000	<0.13	500	**>170**	**>212**	[27]
5-FU	15.3	100.3	186.3	92.7	98.7	1000	1000	1000	1000	11.7	14.2	12.9	**>10**	**0.1**	[27]
DOC	<0.03	<0.03	<0.03	<0.03	<0.03	70	100	91	87	0.12	0.03	0.08	**>2708**	**>2.4**	[27]
MEL	114.0	29.0	18.3	19.0	45.1	153	197	170	173	13.5	18.7	16.1	**3.8**	**0.4**	[27]
Gefitinib	18.0	22.3	15.7	13.7	17.4	58	68	83	70	3.5	4.1	3.8	**4.0**	**0.2**	[27]
**Exp. 2 Chromone derivatives:**															
**A**	2.1	1.0	3.6	1.2	2.0	67	74	272	138	19	>800	>410	**69.0**	**>205**	[30]
**B**	3.2	11.3			7.3	>400	>400		>400	3.8	3.3	3.6	**>55.2**	**0.5**	[32]
**C**	3.5	1.5	5.5	8.3	4.7	400	41	400	280	>400	>400	>400	**59.9**	**>85.1**	[33]
**D**	1.6	1.3			1.5	36	35		36				**24.1**		[34]
**E**	46.0	20.0	36.7	26.3	32.3	390	>400	>400	>397	>400		>400	**>12.3**	**>12.4**	[36]
**F**	9.1	6.0	3.7	3.1	5.5	244	>400	>400	>348	356		355.7	**63.4**	**65.2**	[35]
**G**	13.8		27.7	70.1	37.2	185	273	324	261				**7.0**		[37]
**H**	4.7	5.3			5.0	247	233		240	20		20	**47.8**	**4.1**	[38]

The mean of 50% cytotoxic concentration (CC_50_) of each test compound for human oral squamous cell carcinoma (OSCC) cell lines (Ca9-22, HSC-2, HSC-3, HSC-4) (A) and human normal oral mesenchymal cells (human gingival fibroblast HGF, human periodontal ligament fibroblast HPLF, human pulp cell HPC) (B) and human normal oral epithelial cells (HOK, HGEP) (C) were determined after incubation for 48 h with various concentrations of them. Tumor-specificity (TS) for mesenchymal normal cells *vs* OSCC cells (TS_M_) and that for epithelial normal cells *vs* OSCC (TS_E_) was determined by the following equation: TS_M_ = B/A, TS_E_ = C/A. Structures of A–H (Exp. 2) that showed the highest tumor-specificity in each group are shown in Figure 4. A, (*E*)-3-(4-hydroxystyryl)-6-methoxy-4*H*-chromen-4-one; B, (3*E*)-2,3-dihydro-3-[(3,4-dihydroxyphenyl)methylene]-7-methoxy-4*H*-1-benzopyran-4-one; C, (*E*)-3-(4-cholorostyryl)-7-methoxy-2*H*-chromene; D, 2-(1*H*-indol-1-yl)-6-methoxy-4*H*-1-benzopyran-4-one; E, 2-(4-phenyl-1-piperazinyl)-4*H*-1-benzopyran-4-one; F, 7-methoxy-2-(4-morpholinyl)-4*H*-1-benzopyran-4-one; G, (2*R*,3*aR*,9*aR*)-rac-3*a*,9*a*-dihydro-7-methoxy-4-oxo-2-(2-phenylethenyl)-4*H*-furo[2,3-*b*][1]benzopyran-3,3(2*H*)-dicarboxylic acid 3,3-dimethyl ester; H, 8-chloro-4,4a-dihydro-3-methoxy-3-methyl-3*H*,10*H*-pyrano[4,3-*b*][1]benzopyran-10-one.

**Table 2 medicines-06-00004-t002:** Top six chemical descriptors that showed the highest correlation to cytotoxicity to tumor cells (**T**) or normal cells (**N**) or tumor-specificity (**T–N**). Descriptors are explained in the footnote. Pink, molecular size; yellow, 3D shape; orange, topological shape; blue, electrostatic; green, lipophilicity.

	Category	Number of Descriptors Searched	T	N	T-N	Ref.
A	3-Styrylchromones		OMe at R1	vsurf_DD23	OH at R3	[30]
			OH at R3	G1u	vsurf_DD23	
					G2u	
B	3-Benzylidenechromanones	3134	RDF095i	Mor03v	Mor3m	[32]
			RDF095u	Mor03m	Mor03v	
			RDF095e	Mor09m	SpMAD_AEA(dm)	
			vsurf_IW6	G1u	vsurf_HB7	
			vsurf_ID7	Mor03p	R3m+	
			vsurf_ID1	R3m+	Mor25v	
C	3-Styryl-2*H*-chromenes	330	chi1v	std_dim2	std_dim3	[33]
			KierFlex	E_tor	BCUT_SLOGP_1	
			KierA1	E_oop	vsurf_D4	
			SMR_VSA7	std_dim3	vsurf_R	
			KierA3	vsurf_A	vsurf_D5	
			Weight	BCUT_SMR_1	E-oop	
D	2-Azolylchromones	3062	G3m	SpMin8_Bh(s)	Kp	[34]
			G3e	Q_RPC-	P1p	
			G3v	G3s	Mor32i	
			Gm	G3e	P2p	
			G3p	G3m	Mor32u	
			G3s	Gm	CATS2D_02_LL	
E	3-(*N*-Cyclicamino)chromones	3096	RDF075v	Mor28s	CATS3D_12_LL	[36]
			RDF075p	CATS3D_02_AL	VE3sign_G	
			Mor06s	CATS2D_02_AL	J_D/Dt	
			SpMAD_AEA(dm)	Inflammat-80	FCASA-	
			RDF090p	Depressant-80	CATS3D_11_LL	
			E3m	TDB05i	Chi_G/D	
F	2-(*N*-Cyclicamino)chromones	3089	SpPosA_B(m)	Mor32u	Mor22m	[35]
			SpPosA_B(e)	Mor32e	GCUT_SLOGP_1	
			GCUT_SLOGP_1	VR2_G/D	Mor17v	
			Mor17v	JGI4	Mor17m	
			Mor17m	VR2_G		
			VE1sign_B(v)	SPH		
G	Furo[2,3-*b*]chromones	2820	b_double	rsynth	b_double	[37]
			SlogP_VSA2	b_double	SlogP_VSA2	
			rsynth	SlogP_VSA2	rsynth	
			std_dim3	std_dim3	std_dim3	
			E_str	E_str	b_rotR	
			dens	dens	E_str	
H	Pyrano[4,3-*b*]chromones	3072	R8s	R6v+	R8s	[38]
			J_G	R1s	HATS7i	
			RDF055s	R4v	HATS3i	
			R7s	J_G	HATS3u	
			HATS7s	R4p	HATS7u	
			RTs	R3v+	Mor10i	

b_double: Number of double bonds. Aromatic bonds are not considered to be double bonds. b_rotR: Fraction of rotatable bonds; CATS2D_02_AL: CATS2D Acceptor-lipophilic at lag 02; CATS2D_02_LL: CATS2D Lipophilic-Lipophilic at lag 02; CATS3D_02_AL: CATS3D Acceptor-lipophilic BIN 02 (2.000-3.000Å); CATS3D_11_LL: CATS3D lipophilic-lipophilic BIN 11 (11.000-12.000Å); CATS3D_12_LL: CATS3D Lipophilic-Lipophilic BIN 12 (12.000-13.000Å); Chi_G/D: Randic-like index from distance/distance matrix; chi1v: atomic valence connectivity index; dens: Mass density: molecular weight divided by van der Waal’s volume; Depressant-80: Ghose-Viswanadhan-Wendoloski antidepressant-like index at 80%; E3m: 3rd component accessibility directional WHIM index/weighted by mass; E_oop: out-of-plane potential energy; E_str: Bond stretch potential energy; E_tor: torsion potential energy; FCASA-: Fractional CASA-(negative charge weighted surface area, ASA-times max { qi<0 }) calculated as CASA-/accessible surface area; GCUT_SLOGP_1: The GCUT descriptors using atomic contribution to logP (using the Wildman and Crippen SlogP method); Gm: total symmetry index/weighted by mass; G1u: (the first component symmetry directional WHIM index/unweighted encoding molecular symmetry that extracts the global symmetry information; G2u: (the second component symmetry directional WHIM index/unweighted encoding molecular symmetry that extracts the global symmetry information; G3e: 3rd component symmetry directional WHIM index/weighted by Sanderson electronegativity; G3m: 3rd component symmetry directional WHIM index/weighted by mass; G3p: 3rd component symmetry directional WHIM index/weighted by polarizability; G3s: 3rd component symmetry directional WHIM index/weighted by I-state; G3v: 3rd component symmetry directional WHIM index/weighted by van der Waals volume; HATS3i: Leverage-weighted autocorrelation of lag 3/weighted by ionization potential; HATS3u: Leverage-weighted autocorrelation of lag 3/unweighted; HATS7i: Leverage-weighted autocorrelation of lag 7/weighted by ionization potential; HATS7s: Leverage-weighted autocorrelation of lag 7/weighted by I-state; HATS7u: Leverage-weighted autocorrelation of lag 7/unweighted; Inflammat-80: Ghose-Viswanadhan-Wendoloski anti-inflammatory-like index at 80%; J_D/Dt: Balaban-like index from distance/detour matrix; J_G: Balaban-like index from geometrical matrix; JGI4: Mean topological charge index of order 4; KierA1: First alpha modified shape index; KierA3: Third alpha modified shape index; KierFlex: Kier molecular flexibility index; Kp: K global shape index/weighted by polarizability; Mor03m: signal 03/weighted by mass; Mor03p: signal 03/weighted by polarizability; Mor03v: signal 03/weighted by van der Waals volume; Mor06s: Signal 06/weighted by I-state; Mor09m: signal 09/weighted by mass; Mor10i: Signal 10/weighted by ionization potential; Mor17m: Signal 17/weighted by mass; Mor17v: Signal 17/weighted by van der Waals volume; Mor22m: Signal 22/weighted by mass; Mor25v: signal 25/weighted by van der Waals volume; Mor28s: Signal 28/weighted by I-state; Mor32e: Signal 32/weighted by Sanderson electronegativity; Mor32i: signal 32/weighted by ionization potential in 3D-MoRSE descriptors; Mor32u: signal 32/unweighted in 3D-MoRSE descriptors; OMe at R1: methoxy substitution at the 6-position on the chromone ring group; OH at R3: 4′-hydroxy substitution in the phenyl group of styryl moiety; P1p: 1st component shape directional WHIM index/weighted by polarizability; P2p: 2nd component shape directional WHIM index/weighted by polarizability; Q_RPC-: Relative negative partial charge: the smallest negative partial charge atom i divided by the sum of the negative partial charge atom i; RDF055s: Radial Distribution Function- 055/weighted by I-state; RDF075p: Radial distribution function-075/weighted by polarizability; RDF075v: Radial distribution function-075/weighted by van der Waal’s volume RDF; RDF090p: Radial distribution function-090/weighted by polarizability; RDF095i: Radial Distribution Function - 095/weighted by ionization potential; RDF095u: Radial Distribution Function - 095/unweighted; RDF095e: Radial Distribution Function - 095/weighted by Sanderson electronegativity; rsynth: The synthetic reasonableness or feasibility, of the chemical structure; RTs: R total index/weighted by I-state; R1s: R autocorrelation of lag 1/weighted by I-state; R3m+: R maximal autocorrelation of lag 3/weighted by mass; R3v+: R maximal autocorrelation of lag 3/weighted by van der Waals volume; R4p: R autocorrelation of lag 4/weighted by polarizability; R4v: R autocorrelation of lag 4/weighted by van der Waals volume; R6v+: R maximal autocorrelation of lag 6/weighted by van der Waals volume; R7s: R autocorrelation of lag 7/weighted by I-state; R8s: R autocorrelation of lag 8/weighted by I-state; SCUT_SLOGP_1: using atomic contribution to logP1; SCUt_SMR_1: using atomic contribution to molar refractivity1; SlogP_VSA2: Sum of approximate accessible van der Waal’s surface area i such that logP for atom i is from −0.2 to 0; SMR_VSA7: sum of vi such that Ri > 0.56; SPH: Spherosity; SpMAD_AEA: Spectral mean absolute deviation from augmented edge adjacency matrix weighted by dipole moment edge adjacency indices; SpPosA_B(e): Normalized spectral positive sum from Burden matrix weighted by Sanderson electronegativity; SpPosA_B(m): Normalized spectral positive sum from Burden matrix weighted by mass; SpMin8_Bh(s): Smallest eigenvalue n. 8 of Burden matrix weighted by I-state; std_dim2, std_dim3: standard dimension 2 or 3 that depend on the structure connectivity and conformation; TDB05i: 3D Topological distance based descriptors-lag 5 weighted by ionization potential; VE1sign_B(v): Coefficient sum of the last eigenvector from Burden matrix weighted by van der Waals volume; VE3sign_G: logarithmic coefficient sum of the last eigenvector from geometrical matrix; VR2_G: Normalized Randic-like eigenvector-based index from geometrical matrix; VR2_G/D: Normalized Randic-like eigenvector-based index from distance/distance matrix; vsurf_A: amphiphilic moment; vsurf_D4: hydrophobic volume 4; vsurf_D5: hydrophobic volume 5; vsurf_DD23: the interaction with hydrophobic probe assumed surrounding the molecule; vsurf_HB7: H-bond donor capacity 7; vsurf_ID1: Hydrophobic interaction-energy moment 1; vsurf_IW6: Hydrophilic interaction-energy moment 6; vsurf_ID7: Hydrophobic interaction-energy moment 7; vsurf_R: surface rugosity; Weight: molecular weight.

**Table 3 medicines-06-00004-t003:** Anti-HIV activity of natural products.

Samples	Anti-HIV activity (SI)	Ref.
**Lignin-carbohydrate complex**		
Pine cone of *Pinus parviflora* Sieb. et Zucc	14	[77]
Pine cone of *Pinus elliottii* var. Elliottii	28	[78]
Pine seed shell of *Pinus parviflora* Sieb. et Zucc	12	[79]
Bark of *Erythroxylum catuaba* Arr. Cam.	43	[80]
Husk of cacao beans of Theobroma	311	[81]
Mass of cacao beans of Theobroma	46	[82]
*Lentinus edodes* mycelia extract (L·E·M)	94	[83]
Precipitating fiber fraction of mulberry juice	7	[84]
Dehydrogenation polymers of phenylpropenoids (n = 23)	105	[85]
**Polysaccharides**		
Neutral polysaccharides of pine cone of *P. parviflora* Sieb. et Zucc	1	[86]
Uronic acid-containing polysaccharides of pine cone	1	[86]
**Lower molecular weight polyphenols**		
Hydrolysable tannins (monomer) (MW: 484–1255) (n = 21)	1.8 ± 2.8	[87]
Hydrolysable tannins (dimer) (MW: 1571–2282) (n = 39)	2.3 ± 3.2	[87]
Hydrolysable tannins (trimer) (MW: 2354–2658) (n = 4)	3.4 ± 3.7	[87]
Hydrolysable tannins (tetramer) (MW: 3138–3745) (n = 3)	7.3 ± 6.5	[87]
Condensed tannins (MW: 290–1764) (n = 8)	1.1 ± 0.4	[87]
Flavonoids (MW: 84–648) (n = 92)	1.5 ± 1. 9	[88]
**Herb extracts**		
Green tea leaves Hot water extraction		[89]
Alkaline extraction	3	
Oolong tea leaves Hot water extraction	<0.033	[89]
Alkaline extraction	13	
Orange flower Hot water extraction	<0.5	[89]
Alkaline extraction	>15	
Licorice root Hot water extraction	4	[90]
Alkaline extraction	42	
Alkaline extract of leaves of *Sasa* sp.	86	[86]
Kampo medicines (n = 10)	1.0 ± 0.0	[91]
Constituent plant extracts of Kampo medicines (n = 25)	1.3 ± 0.8	[91]
**Chromones**		
(*E*)-3-(4-Hydroxystyryl)- 6-methoxy-4*H*-chromen-4-one	<1	[30]
(*E*)-3-(4-Chlorostyryl)-7-methoxy-2*H*-chromene	<1	[30]
**Positive Controls**		
Dextran sulfate (molecular mass, 5 kDa)	2956	
Curdlan sulfate (molecular mass, 79 kDa)	11718	
Azidothymidine	23261	
2′,3′-Dideoxycytidine (ddC)	2974	

**Table 4 medicines-06-00004-t004:** SE shows prominent anti-HIV, anti-UV, anti-inflammation and neuroprotective activities.

Samples	Anti-HIV	Anti-UV	Anti-Inflammation	Neuroprotection
*(Target cells)*	(T-cell leukemia)	(HSC-2)	(HPLF)	(Differentiated PC12)
Evaluated by	CC_50_/EC_50_ (+HIV)	CC_50_/EC_50_(+UV)	CC_50_/EC_50_(+IL-1ꞵ)	CC_50_/EC_50_(+Aꞵ_25-35_)
SE	86	38.5	>96.8	56.8
Curcumin		<1.0	1.5	17.3
Gallic acid	<1.0	5.4	0.9	
Ferulic acid	<1.0		>2.9	
*p*-Coumaric acid	<1.0		>3.1	
EGCG	<1.0	7.7		10.7
Resveratrol		<1.0		<1.0
Rikkosan	<1.0	24.1	>4.3	
Hangesyashinto	<1.0	>4.9	285	
Glycyrrhiza	<1.0	4.3	59	
Ref.	[84]	[97,98]	[99]	[100]

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
