# Peer review of "Recent Progress of Basic Studies of Natural Products and Their Dental Application"

_medicines, 2018, doi:10.3390/medicines6010004_

Reviewer 1 Report

It is a well written paper. I would like some more clarity to be introduced in the paper regarding the product application in the management of dental and periodontal disease

Author Response

Reply to Reviewer 1

It is a well written paper. I would like some more clarity to be introduced in the paper regarding the product application in the management of dental and periodontal disease

Thank you for evaluating our paper. 

We added new information of catechin gel in section 3.3.

We newly inserted new section of 4. Lignin-carbohydrate complex (LCC) as anti-HIV resources of the natural kingdom, since one of the main components of SE is LCC, and therefore these show similar biological activity.

We added more information of how to treat the lichenoid dysplasia patient by SE in section of 5.2.

We added more information of how to manufacture the SE containing toothpaste (SETP).

Instead, according to the comments by reviewer 3, we have shortened the description of unpublished clinical data in section 5.5. (other unpublished case reports)

We have stated that “However, previous studies have not yet investigated whether traditional medicines and dietary polyphenols inhibit the periodontitis through blocking the AT1” (Line 656, section 7.3), to connect the last paragraph “This review suggests the efficacy of GTF inhibitors and ARBs to prevent the biofilm formation and periodontitis, respectively.  It is crucial to search for these inhibitors and blockers from the natural kingdom and elucidate their action mechanism” (Line 685, Section 8).

Thank you again for your nice comment.

We hope such modification of the manuscript meets your criticisms.

Reviewer 2 Report

Much of the information provided in this review are the authors' own research and case reports, some of which are only loosely connected.  Publication is recommended however as the review does bring together the collective works of these researchers.

Although I had no problem understanding the text, the manuscript would benefit from a proofreading by a native speaker of English.

There are very many abbreviations throughout this manuscript.  In order to help readers, a list of abbreviations should be provided.

In Table 2, most readers will not know what these descriptors mean.  The authors should provide a definition of each of these.  Maybe add a footnote to this table to "...see Appendix X for definitions of the chemical descriptors."

There are several places where the names of these bacterial genera should be italicized (lines 149, 153, 161, 162, 224)

Line 240:  The “14” of 14C-glucose should be superscript.

Author Response

Reply to Reviewer 2

Much of the information provided in this review are the authors' own research and case reports, some of which are only loosely connected.  Publication is recommended however as the review does bring together the collective works of these researchers.

Although I had no problem understanding the text, the manuscript would benefit from a proofreading by a native speaker of English.

Thank you for recommending our review article for publication.  We have re-checked the English in the text.

There are very many abbreviations throughout this manuscript.  In order to help readers, a list of abbreviations should be provided.

Thank you for good advice.  We have added a list of abbreviations between the section 8 (future direction) and references.

In Table 2, most readers will not know what these descriptors mean.  The authors should provide a definition of each of these.  Maybe add a footnote to this table to "...see Appendix X for definitions of the chemical descriptors."

Thank you for good advice.  We added a list of explanation of chemical descriptors in an alphabetical order in the footnote of Table 2.

There are several places where the names of these bacterial genera should be italicized (lines 149, 153, 161, 162, 224)

Thank you for pointing out our typographical errors.  We have corrected them into italicized form.

Line 240:  The “14” of 14C-glucose should be superscript.

Thank you for your careful check again.     We have corrected “14C” to “14C”

Thank you again for your nice comment.  We hope such modification of the manuscript meets your criticisms.

Reviewer 3 Report

Review of Recent progress of basic studies of natural products and their dental application

In general:

All claims needs to be referenced. There are numerous claims in this article that needs proper citations.

The authors are very broad in their claims and grossly oversimplify large topics. The studies shown are small and of little value. I would not recommend this publication to be accepted.

Detailed comments:

Abstract: Very broad and general text. Very few specific numbers. Focus on three groups of natural products, the chromones, the polyphenols and the ligning carbohydrate complexes. Then there were some traditional herbal remedies with highly suggestive applications.

Section 1:

Line 47-52: The first full paragraph of the introduction had no references, yet it contained many claims.

Line 47-52: The topics oral inflammation, stomatitis and oral diseases are very broad, may have several different (and excluding) causes and to treat this as one concept is to stretch the topic.

Line 53-62: This section spans a lot of different polyphenols, yet contain only a single reference to lignin carbohydrate complex(es). This must be documented better.

Line 65-70: While it is generally true what is listed here, to me, this seems like a large oversimplification. Hygiene is not mentioned. Nutrition is not mentioned. Fluoride is not mentioned. And orally administered products may or may not be able to cause any biochemical effect on the cells. There are several obstacles for a compound, crude or pure, needs to overcome in order to cause an effect.

Section 2:  

Line 73-74. Please reference this with either a statistic or some other documentation showing that this is the case.

Line 74-76. This is completely normal. As long as a new treatment is better than the existing one, even if only by a few percent, it should be used. It is unethical to withhold a better treatment. This claim made by the authors indicate a low level of understanding of anticancer drug discovery and development.

Table 1: What does “TS” stand for? It is not explained anywhere in the text.

What are the numbers in table 1 representing? IC50 values for the compound(s) versus the cell line?

Most, if not all, anticancer drugs show cell selectivity to a certain degree. The drugs are toxic. They are designed to be toxic. Some have low therapeutic index, but they are still the best option in some cases. Many compounds show great anticancer activity in vitro, however, that does not mean they can be used as drugs for humans. Many of the drugs listed in table 1 are on the WHOs list of essential drugs, a list consisting of the most effective and safe medications needed in human healthcare.

Figure 2: Which concentration is used here? Doxorubicin is toxic to nearly all cell lines if the dose is high enough.

Figure 3 elegantly shows the therapeutic index of some listed compounds. However, there are several improvements needed. Firstly, “normal cells” can mean many things. The various groups of compounds could be expanded, as “amides”, “ketones”, “antioxidants”, etc does not give away much information about which compounds they are.

The “poly-herbal formula (n=1)” seems to indicate that this is a perfect anticancer drug. What is this compound/extract? This needs to be more precise, because without precision, the whole point of figure 3 falls.

Figure 4: What types of control compounds were used in this study. Are these compounds better or worse than current anticancer agents? What are the IC50 values of these? These numbers are essential in order to evaluate the true usefulness of these compounds. How does the compounds work?

Figure 5: Which compound is 5c?

Table 2: You conclude that the structure is the most important factor. This is not surprising. What is the target these compounds act upon? If you would like to do computer assisted drug development, I would include molecular modeling.

Section 3:

Lines 135-141: Again, no reference(s) in the first paragraph of this section.

Figure 6 is cited after the figure is shown. It should be the other way around.

Line 166: Figure 5 or figure 6?

Section 3.3

Line 221: The first sentence does not give any information. Any change in a chemical system might affect the equilibrium. Also, there is no reference.

Line 230: “low-molecular polyphenol from plant origin” can mean many things. Could you give some examples of structures or complexes. Which chemical weights do you consider low-molecular?

Figure 9 and the rest of section 3.3. Are these observed effects dose-dependent?

Line 243-247: Claims needs references.

Line 250-257: The availability of this remedy is country specific. This is not something that is available at my local pharmacy. Again, this is an example of a gross oversimplification in this review article.

Line 260-266: This describes one experiment in one patient. There are no ways to draw useful conclusion from this single data point. There is also no information (in the text) about what controls are factored into this experiment.

Line 292-293: Claims such as “However, the incidences of such shunt troubles were apparently reduced by SE treatment (D) and questionnaire on the clinical effects of SE showed the good outcome (E)” and “one patient (84 years old) still receives the clinical trial, with little or no trouble of the shunt, possibly due to the preservation of blood fluidity and maintenance of blood vessel possibly by iron chlorophyllin and antioxidants in SE [61].” Are dangerous to conclude from. The sample size is very little and the apparent effect is hard to measure directly.

Section 4,4 and 4,5 also describe short or small human trials. While the first trial had some objective measurements, the latter is only one person and a very subjective end point to analyze. The value of such a study is very low.

Line 321-323 is downright not scientific. “It is said that…” without any references is meaningless.

The results from this study is also of little value as it is very hard to draw meaningful data without proper controls.

Line 333: Where is this case reported? How was the study performed? Which factors were evaluated?

Author Response

Reply to Reviewer 3

In general:   All claims needs to be referenced. There are numerous claims in this article that needs proper citations.  The authors are very broad in their claims and grossly oversimplify large topics. The studies shown are small and of little value. I would not recommend this publication to be accepted.

Thank you for your careful check to improve our manuscript.  We have corrected the text, and added many of proper citations, as indicated by red.

Detailed comments:

Abstract: Very broad and general text. Very few specific numbers. Focus on three groups of natural products, the chromones, the polyphenols and the ligning carbohydrate complexes. Then there were some traditional herbal remedies with highly suggestive applications.

Thank you for your good suggestion.   According to your advice, we lined up the abstract in the order you have recommended, as follows.  I hope this modification make the storyline more clear, and meets your criticism.

Abstract:  The present article reviews the research progress of three major polyphenols (tannins, flavonoids and lignin carbohydrate complexes], chromone (backbone structure of flavonoids) and herbal extracts. Chemical modified chromone derivatives showed highly specific toxicity against human oral squamous cell carcinoma cell lines, with much lower toxicity against human oral keratinocytes, as compared with various anticancer drugs.  QSAR analysis suggests the possible correlation between their tumor-specificity and three-dimensional molecular shape.  Condensed tannins in the tea extracts inactivated the glucosyltransferase enzymes, involved in the biofilm formation. Lignin-carbohydrate complexes (prepared by alkaline extraction and acid-precipitation) and crude alkaline extract of the leaves of Sasa sp. (SE, available as an over-the-counter drug) showed much higher anti-HIV activity, than tannins, flavonoids and Japanese traditional medicine (Kampo). Long-term treatment with SE and Kampo showed an anti-inflammatory and anti-oxidant effects in small size of clinical trials. Although the anti-periodontitis activity of synthetic angiotensin II blockers has been suggested in many papers, natural angiotensin II blockers has not yet been tested for their possible anti-periodontitis activity. There should be still many unknown substances that are useful for treating the oral diseases in the natural kingdom.  

Section 1:

Line 47-52: The first full paragraph of the introduction had no references, yet it contained many claims.

We have added new references about the etiology [1, 2and treatment [3], and rewrote this section.

Line 47-52: The topics oral inflammation, stomatitis and oral diseases are very broad, may have several different (and excluding) causes and to treat this as one concept is to stretch the topic.

We rewrote this paragraph as follows:

The etiology of stomatitis is largely unclear [1]. However, oral inflammation such as stomatitis are considered to be triggered or aggravated by various factors including bacterial and viral infections, nutritional deficiencies, declined immune functions, allergic reactions, radiotherapy, stress, cigarettes, diseases and genetic backgrounds [1, 2].  Applications of topical steroids, transdermal patches, vitamins, throat lozenges, mouth washes, and cryotherapy are sometimes not effective for the treatment of stomatitis, and therefore exploration of new-type of treatment are necessary [3]. In this sense, natural products having broader spectrum of biological activities are potential candidates as alternative medicine for oral diseases.

Line 53-62: This section spans a lot of different polyphenols, yet contain only a single reference to lignin carbohydrate complex(es). This must be documented better.

I have added new references that described the classification and characteristic structures of tannins, flavonoids and lignin-carbohydrate complex as new references [5, 7, 8, 9].

Line 65-70: While it is generally true what is listed here, to me, this seems like a large oversimplification. Hygiene is not mentioned. Nutrition is not mentioned. Fluoride is not mentioned. And orally administered products may or may not be able to cause any biochemical effect on the cells. There are several obstacles for a compound, crude or pure, needs to overcome in order to cause an effect.

We have inserted three references about the oral hygiene [12], food intake [13] and fluoride [14].

We have added the conditions that “if they have a chance to bind the target molecules or pattern-recognition receptors such as TLR2, TLR4, Dectin-1 (receptor for glucan), and Dectin-2 (receptor for LCC or mannan) in keratinocytes, macrophages, monocytes and dendritic cells [16].”  Line 89-82.

Section 2: 

Line 73-74. Please reference this with either a statistic or some other documentation showing that this is the case.

We have added three references [17-19]. 

Line 74-76. This is completely normal. As long as a new treatment is better than the existing one, even if only by a few percent, it should be used. It is unethical to withhold a better treatment. This claim made by the authors indicate a low level of understanding of anticancer drug discovery and development.

We have corrected the sentences, by incorporating three new references: 

However, the incidence of complete response in gastroesophageal cancer (GEC) patients treated with targeted agents has been reported to be 2.0%, only 0.3 increase from the control arms [17]. ErbB receptor-targeting inhibitors failed to show any significant differences on overall response rate, clinical benefit rate, and overall survival, with the increased risk of serious adverse events [18].  Likewise, cyclin-dependent kinase inhibitor combined with chemotherapy slightly increased the mean progression-free survival, but also stimulated the senescent-associated marker expression by yet unknown mechanism [19]. This points out another unfavorable effect of targeted therapy, the resolution of which we have to find urgently. 

Administration of anticancer agents has been reported to induce skin toxicity [20-26]. This prompted us to re-evaluate the cytotoxicity and tumor-specificity of anticancer drugs. We demonstrated for the first time that classical anticancer drugs (doxorubicin, daunorubicin, etoposide, mitomycin C, methotrexate, 5-fluorouracil, melphalan), and molecular targeted therapeutic drug (gefinitib) are highly toxic to epithelial normal cells (keratinocytes) (TSE = 0.1~1.5), usually one to two-orders higher than that against mesenchymal normal cells (TSM=3.8-92.9) [27] (Exp. 1, Table 1).

Table 1: What does “TS” stand for? It is not explained anywhere in the text.

What are the numbers in table 1 representing? IC50 values for the compound(s) versus the cell line?

We have added the CC50 values for all four human oral squamous cell carcinoma cells lines (Ca9-22, HSC-2. HSC-3, HSC-4), three human normal oral mesenchymal cells (HGF, HPLF, HPC) and two human normal oral epithelial cells (HOK, HGEP) and how to calculate the tumor specificity (TSM for comparison between OSCC vs normal mesenchymal cells, TSE for OSCC vs normal epithelia cells) in the foot note of Table 1.  In the footnote, we have added the full names of eight chromones A~H that showed the highest TS activity in each group.

Most, if not all, anticancer drugs show cell selectivity to a certain degree. The drugs are toxic. They are designed to be toxic. Some have low therapeutic index, but they are still the best option in some cases. Many compounds show great anticancer activity in vitro, however, that does not mean they can be used as drugs for humans. Many of the drugs listed in table 1 are on the WHOs list of essential drugs, a list consisting of the most effective and safe medications needed in human healthcare.

We have cited seven papers [20-26] that demonstrated the skin toxicity induced by anticancer agents.  This stimulated us to re-evaluate the cytotoxicity and tumor-specificity of anticancer drugs.

Figure 2: Which concentration is used here? Doxorubicin is toxic to nearly all cell lines if the dose is high enough.

3A

We have added the concentration of doxorubicin (10 μM) in Figure 3A.  We added this information in Figure 2. 

Figure 3 elegantly shows the therapeutic index of some listed compounds. However, there are several improvements needed. Firstly, “normal cells” can mean many things. The various groups of compounds could be expanded, as “amides”, “ketones”, “antioxidants”, etc does not give away much information about which compounds they are.

We explained the tumor and normal cell in the legend of Figure 3.  More detailed information of groups of amides, ketones and antioxidants are now shown in Figure 3.

The “poly-herbal formula (n=1)” seems to indicate that this is a perfect anticancer drug. What is this compound/extract? This needs to be more precise, because without precision, the whole point of figure 3 falls.

We added the following paper as [28] 

Published in Anticancer Res. 2002 Mar-Apr;22(2B):1217-23. PMID: 12168929

Miyamoto M, Sakagami, Minagawa K, Kikuchi H, Nishikawa H, Satoh K, Komatsu N, Fujimaki M, Nakashima H, Gupta M, Sarma DNK and Mitra SK:  Tumor-specificity and radical scavenging activity of poly-herbal formula.  Anticancer Res 22: 1217-1224, 2002.

Abstract A total of 14 poly-herbal formula extracts were compared for their biological activities both in vivo and in vitro. Pretreatment of mice with the extracts protected them from E. coli infection to various extents. Among the extracts, the HD-12 and DLH-3073 extracts showed the highest cytotoxicity against both HIV-infected and mock-infected MT4 cells, without induction of any apparent anti-HIV activity. The extracts showed significantly higher cytotoxic activity against five human tumor cell lines (HSC-2, HSC-3, HSG, MT-4, HL-60) than against three normal human cell lines (HGF, HPC, HPLF). Agarose gel electrophoresis demonstrated that the HD-12 and DLH-3073 extracts induced intemucleosomal DNA fragmentation in HL-60 cells. ESR spectroscopy showed that all the extracts produced radicals and this was paralleled by their ability to scavenge the superoxide anion (produced by hypoxanthine-xanthine oxidase reaction), the hydroxyl radical (produced by Fenton reaction) and nitric oxide (produced by NOC- 7) in the presence of radical trapping agents. Higher and lower concentrations of extracts enhanced or reduced respectively, the radical intensity of sodium ascorbate, suggesting their bimodal actions. The tumor specificity and antioxidant properties of the herb extracts further suggest their medicinal efficacy.

A total of 14 poly-herbal formula extracts were obtained from Dr. Mitra SK, Himalaya drug company. Bangalore, India.  Since then, our collaboration study has been stopped, and therefore we could not provide more information.  However, if we can do such collaboration, we would like to do so.    At this moment, we can only state that “The active principle (s) are yet to be determined.”   Line 138.

Figure 4: What types of control compounds were used in this study. Are these compounds better or worse than current anticancer agents? What are the IC50 values of these? These numbers are essential in order to evaluate the true usefulness of these compounds. How does the compounds work?

We have used 5-FU, DXR in each of our published papers [30-38].   One of these IC50 values are listed in Table 1.  It is true that absolute value of CC50 is changeable in each experiment, therefore, we have to add these positive control to compare the TS value of each test compounds with that of positive control.

Figure 5: Which compound is 5c?

We rewrote “5c” to “7-Methoxy-2-(4-morpholinyl)-4H-1-benzopyran-4-one (Compound F, Figure 4F)”  The concentration of actinomycin D (Act. D) was also added.

Table 2: You conclude that the structure is the most important factor. This is not surprising. What is the target these compounds act upon? If you would like to do computer assisted drug development, I would include molecular modeling.

We have added the following sentences:

“For example, we have reported previously that T – N of 3-styrylchromones can be estimated by diameter (largest value in the distance matrix defined by the elements Dij), vsurf_DD23 and R3 OH (n=15, R2=0.764, Q2=0.570, s=0.308) (right), according to the following equation: T – N =0.607(±0.169)diameter – 0.121 (±0.035)vsurf_DD23 + 1.11 (±0.235)R3OH – 7.17 (±2.26) [30]. QSAR analysis can be applied to estimate the most potent chemical structures. By repeating the process of synthesis of the estimated structure and reconfirmation of its activity, more active compounds with defied structure will be manufactured.”  Line 188-194

“Metabolomic analysis is powerful to determine the early event of cell death induction process. We have reported that compound A (which induced apoptosis) increased the intracellular levels of diethanolamine and CDP-choline and reduced that of choline, suggesting the down-regulation of the glycerophospholipid pathway [34]. It remains to be determined what metabolic pathway are affected by compound F (which did not induce apoptosis).”  Line 307-311

Section 3:

Lines 135-141: Again, no reference(s) in the first paragraph of this section.

We have added two new reference [39, 40].  Line 319.   

Figure 6 is cited after the figure is shown. It should be the other way around.

We cited Figure 6 before the figure was shown.

Line 166: Figure 5 or figure 6?

We have corrected Figure 5 to Figure 6.   

Section 3.3

Line 221: The first sentence does not give any information. Any change in a chemical system might affect the equilibrium. Also, there is no reference.

The first sentence was deleted.  

Line 230: “low-molecular polyphenol from plant origin” can mean many things. Could you give some examples of structures or complexes. Which chemical weights do you consider low-molecular?

We have added the names of following compounds:  “hydrolysable tannins (gallotannin, ellagitannin), condensed tannins (proanthocyanin, catachins), complex tannins” in Line 405.

Figure 9 and the rest of section 3.3. Are these observed effects dose-dependent?

We stated that “With the increase of substrate of GTF, the production of glucan reached the plateau (near saturation) level. Even if the substrate concentration is enough, OTF6 effectively inhibited the production of glucan.”  Line 411-412.

Line 243-247: Claims needs references.

We have added new reference [74].  Line 416

Line 250-257: The availability of this remedy is country specific. This is not something that is available at my local pharmacy. Again, this is an example of a gross oversimplification in this review article.

We newly inserted new section of 4. Lignin-carbohydrate complex (LCC) as anti-HIV resources of the natural kingdom with new Table 3, since one of the main components of SE is LCC, and therefore these show similar biological activity.

We explained why we have introduced SE as the representative of alkaline extract by adding the following sentences:

“Although alkaline extracts of plants showed much higher anti-HIV activity than corresponding hot water extracts [86, 89-91], only three papers from other groups have reported the anti-angiogenic and neuroprotective activity of alkaline extracts [92-94].  Also, only two papers have been published on the isolation and fractionation of lignin from bamboo, however, they reported no data of biological activity [95, 96].  Based on these backgrounds, we reviewed mostly our research topics of SE.”  Line 465-469

Line 260-266: This describes one experiment in one patient. There are no ways to draw useful conclusion from this single data point. There is also no information (in the text) about what controls are factored into this experiment.

This study is based on study of one chronic patient of oral lichen planus.  The data of this kind of research is not found in any other journals, and is very much informative.  We have also added more treatment detail, due to the suggestion by reviewer 1.

Line 292-293: Claims such as “However, the incidences of such shunt troubles were apparently reduced by SE treatment (D) and questionnaire on the clinical effects of SE showed the good outcome (E)” and “one patient (84 years old) still receives the clinical trial, with little or no trouble of the shunt, possibly due to the preservation of blood fluidity and maintenance of blood vessel possibly by iron chlorophyllin and antioxidants in SE [61].” Are dangerous to conclude from. The sample size is very little and the apparent effect is hard to measure directly.

We stated that “we did not show any significant fluctuation (measured by SRL Inc.) due to the large variation of the data based on the small numbers of patients”. 

Section 4,4 and 4,5 also describe short or small human trials. While the first trial had some objective measurements, the latter is only one person and a very subjective end point to analyze. The value of such a study is very low.

We have saved the first part, but the latter part was treated as unpublished data in new section of 5.5.  As recommended by reviewer 1, we have supplemented how to manufacture the SE-containing tooth paste. 

“Considering the potent anti-HIV, anti-inflammatory effects of SE, we have manufactured SE containing toothpaste (SETP) for the first time.  The SETP is composed of 26.2% SE, 0.1% isopropylmethylphenol, base materials, cleaning agents, humectants, flavoring substances, sweetening agent, stabilizers, binding agent and washing soap). We have selected the 26.2% SE, since we observed that treatment of periodontal ligament fibroblasts for 1 min with 50% of SE did not affect the cell viability, and approximately 8 ml of saliva were produced and accumulated in the oral cavity by 5 min of tooth brushing. SETP can be obtained at the drug store.”   Line 538-544.

Line 321-323 is downright not scientific. “It is said that…” without any references is meaningless.

We have deleted the unscientific sentences, and the following sentences “It is said that ….. the culprits of maloder”

The results from this study is also of little value as it is very hard to draw meaningful data without proper controls.

Line 333: Where is this case reported? How was the study performed? Which factors were evaluated?

We have moved these parts to the “5.5 Other unpublished case reports”

Thank you again for your careful check of the manuscript.  We hope such modification of the manuscript meets the valuable comments and constructive criticisms raised by you.

Reviewer 4 Report

The manuscript “Recent progress of basic studies of natural products and their dental application” address an interesting topic, worth to be evaluated for the journal “Medicine”.

Unfortunately authors seem to have submitted a paper very below their standard with a very poor writing quality. The entire manuscript is not soundly and, as a reader, I can’t catch the coinceivement of the work.

Single phytochemicals, extracts, medicinal plants and poliherbal formulations appear to be randomly put in the manuscript and they are described in an unclear way, with superfluous figure and schemes.

 In a review on natural products for their dental application, firstly authors must describe the state of the art, clinical trial, in vivo trials, mechanisms elucidated by in vitro studies, in this order; then, if they want to review their previous work (and I wonder why, being a self citing based work not very ethic), at least they can build a solid scientific story, giving an explanatory discussion with limits and perspectives.

Author Response

Reply to Reviewer 4

The manuscript “Recent progress of basic studies of natural products and their dental application” address an interesting topic, worth to be evaluated for the journal “Medicine”.

Thank you for evaluating our review article.

Unfortunately authors seem to have submitted a paper very below their standard with a very poor writing quality. The entire manuscript is not soundly and, as a reader, I can’t catch the coinceivement of the work.

We have reconstructed the manuscript to make the readers to understand well.

Single phytochemicals, extracts, medicinal plants and poliherbal formulations appear to be randomly put in the manuscript and they are described in an unclear way, with superfluous figure and schemes.

We have made the story line of the manuscript more clear, by adding more information (new references, data, appendix, abbreviation) into the text.

 In a review on natural products for their dental application, firstly authors must describe the state of the art, clinical trial, in vivo trials, mechanisms elucidated by in vitro studies, in this order; then, if they want to review their previous work (and I wonder why, being a self citing based work not very ethic), at least they can build a solid scientific story, giving an explanatory discussion with limits and perspectives.

Considering much smaller parts for in vivo and clinical work, as compared with in vitro work, we described in the order of: three groups of natural products, the chromones, the polyphenols and the lignin carbohydrate complexes, and then the application of traditional herbal remedies.  To make the article more logically, we explained why we focused on chromones and SE.   We added new section of 4: Alkaline extract of the leaves of Sasa sp. (SE), that was necessary to continue to the section 5.  We described more clearly how to manufacture the SE containing toothpaste (SETP).  Unpublished clinical data was described in much concise manner in the section of 5.5. Other unpublished case reports.

Thank you again for your nice comments.  We hope such modification of the manuscript meets your criticisms.

Round  2

Reviewer 3 Report

Re-review of “Recent progress of basic studies of natural products and their dental application”

It is good to see the authors have corrected most of the minor concerns I raised in the previous round of review, however, my main point of critique remains. The manuscript is, in my opinion, not a classical review article, but instead a collection of three main projects and some unpublished case studies. The three projects could all be single articles, however, taken together I cannot see a clear link between them other than the “dental application”. One of the case studies involve a skin cream and another deals with unpleasant body odor. They are simple, one person case studies, there are no controls and I think the results are speculative at best.

My recommendation is to publish the results of the individual projects in separate papers as I cannot recommend this review to be published.

Reviewer 4 Report

The manuscript was actually improved and authors corrected many parts. Nevertheless I can't still catch the conceivement of the whole work. My suggestion is to limit the work to a little but specified section, i.e. chromones, adequately described and interesting.